# Pushing Mixture of Experts to the Limit: Extremely Parameter Efficient MoE for Instruction Tuning

**Ted Zadouri[1], Ahmet Üstün[1], Arash Ahmadian[1], Beyza Ermiş[1], Acyr Locatelli[2], Sara Hooker[1]**
[1] Cohere For AI, [2] Cohere
{ted,ahmet,arash,beyza,acyr,sarahooker}@cohere.com

## Abstract

The Mixture of Experts (MoE) is a widely known neural architecture where an ensemble of specialized sub-models optimizes overall performance with a constant computational cost. However, conventional MoEs pose challenges at scale due to the need to store all experts in memory. In this paper, we push MoE to the limit. We propose extremely parameter-efficient MoE by uniquely combining MoE architecture with lightweight experts. Our MoE architecture outperforms standard parameter-efficient fine-tuning (PEFT) methods and is on par with full fine-tuning by only updating the lightweight experts – less than 1% of an 11B parameters model. Furthermore, our method generalizes to unseen tasks as it does not depend on any prior task knowledge. Our research underscores the versatility of the mixture of experts architecture, showcasing its ability to deliver robust performance even when subjected to rigorous parameter constraints. Our code used in all the experiments is publicly available here: https://github.com/for-ai/parameter-efficient-moe.

## 1 Introduction

A conventional training paradigm is to apply the weights of a model to each input. Arguably, this is not efficient since a given input may not need all of a model's capacity. In contrast, MoEs build on the premise that sub-modular components – so called experts – can specialize in different types of inputs. This emphasis on conditional computation has important efficiency side-effects such as constant inference costs. This has made MoEs an area of significant research and widespread adoption in the era of large-scale Transformers where scaling has increased deployment and latency costs (Shazeer et al., 2018; Riquelme et al., 2021; Du et al., 2022; Fedus et al., 2022).

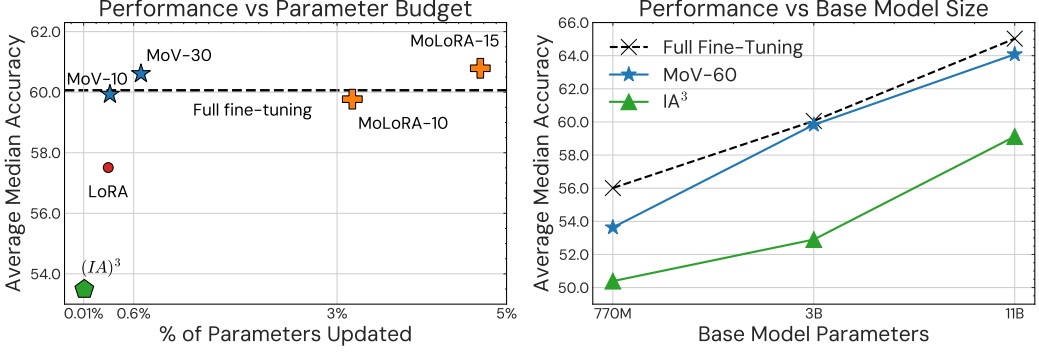

Figure 1: *Left*: Our MoV and MoLORA outperform single PEFT methods using a comparable amount of parameters demonstrated for T5-XL (3B). *Right*: Our MoV approach scales up to 11B; with tiny parameter updates, achieving very competitive performance to full fine-tuning.

While the majority of work to-date has focused on MoEs as a pretraining strategy,the inherent motivation of MoEs is not confined solely to pretraining. In fact, the merits of MoEs are arguably well suited to an *instruction fine-tuning* setting where the data is often deliberately structured to represent a diverse set of tasks, often referred to as multi-task finetuning (Chung et al., 2022; Wei et al., 2022; Sanh et al., 2022; Longpre et al., 2023; Muennighoff et al., 2023).

In this work, we pose the question *can we leverage MoEs for instruction fine-tuning?* One of the main drawbacks of MoEs paradigm is that it introduces an extreme amount of total parameters (Fedus et al., 2022). Despite the conditional computation, fully fine-tuning MoE architecture is computationally demanding given the need to update all the parameters. For most practitioners, given the scale of modern LLMs (Brown et al., 2020; Touvron et al., 2023; Kaplan et al., 2020; Anil et al., 2023) this is an infeasible computational cost.

Thus, we focus on a more realistic setting for everyday practitioners – *can we successfully apply MoEs to parameter-efficient fine-tuning (PEFT)* methods such as $(IA)^3$ (Liu et al., 2022) or LORA (Hu et al., 2021) which only fine-tune a far smaller number of parameters. This is a significant challenge not only since our aim is to update only a small percentage of all parameters but as we also navigate the optimization challenges inherent to MoEs already noted by prior work (Chen et al., 2022) in a more constrained environment.

In this work, we propose a new framework that leverages the benefits of MoE in a severely constrained computational environment. We introduce **Mixture of Vectors (MoV)** and **Mixture of LORA (MoLORA)**, a parameter-efficient adaptation of the Mixture of Experts approach. Unlike the standard MoE, our framework can be utilized in a parameter-limited setting due to its lightweight nature. We evaluated our framework by fine-tuning T5 models (Raffel et al., 2020) ranging from 770M to 11B, using P3 instruction tuning mixture (Sanh et al., 2022) which includes 12 different tasks from 62 datasets. Remarkably, our method achieves performance parity with full fine-tuning on unseen tasks by updating less than 1% of the parameters. It also significantly outperforms base parameter-efficient techniques like $(IA)^3$ or LORA.

**Contributions (1)** We present extremely parameter-efficient MoEs. This architecture leverages MoEs in a more realistic setting using modular and lightweight experts. Our MoEs can be used to fine-tune a dense model by updating less than 1% of its parameters. **(2)** Instruction fine-tuning with our proposed methods consistently outperforms traditional parameter efficient methods on unseen tasks, while maintaining high parameter efficiency across different scales. The mixture of $(IA)^3$ vectors (MoV) achieves up to 14.57% and 8.39% improvements over the standard $(IA)^3$ at 3B and 11B model sizes respectively. This superiority holds across different model sizes, types of experts and trainable parameter budgets. **(3)** We show that our recipe can match the performance of *full fine-tuning* at large scales while updating a tiny fraction of the model parameters. Our results across 8 datasets from unseen tasks show that our MoV which updates just 0.32% and 0.86% of the parameters in the 3B and 11B models achieves *higly competitive* performance to full fine-tuning with a significantly reduced computational cost. **(4)** Finally, we present an extensive set of ablation studies that systematically evaluate the efficacy of various MoE architectures and PEFT strategies at various model sizes, different adapter types, the number of experts, routing mechanisms, and the importance of optimizing hyperparameters, especially given the sensitivity of MoE.

## 2 METHODOLOGY

The standard approach for instruction tuning is to fine-tune all model parameters leading to high compute and memory costs. Our method offers an alternative by pushing the mixture of expert architecture to an extreme degree of parameter efficiency with parameter-efficient fine-tuning (PEFT) methods. In this section, we describe our framework in detail.

### 2.1 PARAMETER-EFFICIENT FINE-TUNING WITH $(IA)^3$ AND LORA ADAPTERS

PEFT methods address the challenges associated with updating a large number of parameters – especially emerging at scale when fully fine-tuning an LLM – by restricting weight updates to a limited number of parameters. To show how our method scales with different PEFT techniques, we

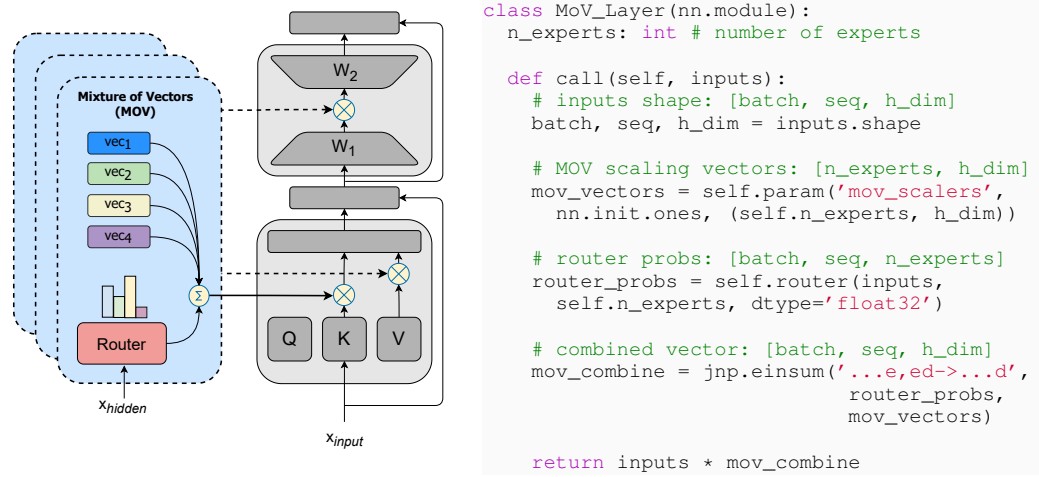

Figure 2: *Left*: Overview of the MoV architecture highlighting soft-merging where only the vectors and router are updated for each multi-head attention block, as denoted by color. *Right*: JAX-like pseudo-code illustrating the core implementation of a MoV layer.

experiment with both (IA)$^3$ and LORA. These methods add a small number of parameters to the existing pre-trained model. We briefly introduce each PEFT method below:

**(IA)$^3$** (Liu et al., 2022) introduces three new vectors, $l_k \in \mathbb{R}^{d_k}$, $l_v \in \mathbb{R}^{d_v}$, $l_{ff} \in \mathbb{R}^{d_{ff}}$ which re-scale key and value activations in self-attention, and intermediate activations in position-wise feed-forward layers:

$$\text{softmax}\left(\frac{Q(l_k \odot K^T)}{\sqrt{d_k}}\right)(l_v \odot V); \quad (l_{ff} \odot \gamma\,(W_1 x))W_2 \quad \quad ((\text{IA})^3)$$

where $Q$, $K$, $V$ are query, key, and value projection matrices for self-attention, and $W_1$, $W_2$ are frozen weights of the feed-forward layers in the pretrained model. Since (IA)$^3$ only updates $l_k$, $l_v$, $l_{ff}$ rescaling vectors for each Transformer layer[1], it is extremely parameter-efficient. For the 3 billion parameter T5 model (Raffel et al., 2020), it only updates $0.018\%$ of the total parameters.

**Low-Rank adaptation** (LORA; Hu et al., 2021) optimizes low-rank decomposition of dense layers in LLMs. For a pre-trained weight matrix $W_0 \in \mathbb{R}^{d_m \times d_p}$ and input activation $x \in \mathbb{R}^{d_m}$, LORA decomposes $W_0$ into two low-rank matrices:

$$h = W_0 x + \Delta W x = W_0 x + BAx \quad \quad (\text{LORA})$$

where $B \in \mathbb{R}^{d_p \times r}$ $A \in \mathbb{R}^{r \times d_m}$, and the rank $r \ll \min(d_m, d_p)$. During fine-tuning, all pretrained weights are frozen, and only $A$ and $B$ matrices are updated.

We use LORA adaptation for all the linear layers in each Transformer block including query $Q$, key $K$, value $V$, output $O$ of the self-attention and the feed-forward layers $W_1$ and $W_2$. To illustrate its parameter efficiency, for a T5 3B model, LORA with a rank of 4, updates $0.3\%$ of the model parameters.

## 2.2 Extremely Parameter Efficient Mixture of Experts

We propose an extremely parameter-efficient Mixture of Experts (MoE) framework that leverages lightweight "adapters" as experts on top of a pretrained dense model. Concretely, the MoE is a family of neural network architecture that enables conditional computation through multiple experts that are activated based on a gating mechanism (router). An MoE layer consists of a router network $R$ and a

---

[1]For an encoder-decoder model with L number of layers in both sides, (IA)$^3$ introduces $L(d_k + d_v + d_{ff})$ new parameters for encoder and $L(2d_k + 2d_v + d_{ff})$ for decoder, due to the additional encoder-decoder attention.

set of $n$ experts $E_1, ..., E_n$ where each expert $E_i$ is a parameterized function. Following Fedus et al. (2022), our router network consists of a dense layer with trainable weights $W_g \in \mathbb{R}^{d_m \times n}$ followed by a *softmax* function which takes an intermediate token representation $x$ as input and combines the output of each expert based on the gating scores $s_1, ..., s_n$:

$$s_i = R(x)_i = \text{softmax}(W_g^T x) \qquad \text{(Router)}$$

$$y = \sum_{i=1}^{n} s_i \cdot E_i(x) \qquad \text{(MoE)}$$

For Transformer models (Vaswani et al., 2023), dense feed-forward layers are replaced by MoE layers where each expert $E_i$ corresponds to an independent dense feed-forward network. This multiplies the total number of model parameters as each expert size and number of experts increases. However, in our parameter-efficient MoE architecture, we replace each expert with a lightweight PEFT adapter such as (IA)$^3$ vectors or LORA adapters. During fine-tuning, pretrained weights of dense layers remain fixed, while experts and router layers are trained from scratch. Unlike the standard MoE, our lightweight experts learn to adapt the pretrained Transformer layers in the fine-tuning time. In this way, our MoE framework requires a limited number of parameter updates and does not introduce a huge model size in total.

In addition to parameter efficiency, (IA)$^3$ vectors enable routing computation with *soft merging* that leads to a reduction in FLOPs without information loss. Concretely, since (IA)$^3$ vectors are used as linear functions, we compute a weighted average of experts first and then apply a PEFT transformation using the combined expert $E_{mix}$ as in Muqeeth et al. (2023):

$$E_{mix} = \sum_{i=1}^{n} s_i \cdot E_i; \ \ y = E_{mix}(x) \qquad \text{(Soft Merging)}$$

For LoRA experts, unlike (IA)$^3$ vectors, we compute a weighted average across *outputs* of LoRA adapters to not lose the memory advantage of low-rank projection.

We call the variants of our method as *Mixture of Vectors* (**MoV**) and *Mixture of LORA* (**MoLORA**) that leverage (IA)$^3$ vectors or LORA adapters as experts respectively, both demonstrating consistent gains over the corresponding PEFT method. Figure 2 shows the architecture of a MoV layer together with the corresponding pseudo-code.[2] Only updating a small fraction of parameters through MoV and MoLORA has multiple practical benefits not only to training but to inference time, with the latter being unique to MoE architectures. We provide a brief overview of these gains below:

**Efficiency in training** Our parameter-efficient MoE formulation leads to a significant reduction in memory. The freezing of most parameters during training reduces the computational overhead of calculating gradients for model parameters but also reduces the memory requirements of storing the optimizer states for the model.

**Efficiency at inference** The inherent structural modularity of our MoV and MoLORA methods allows for significant memory gains at inference time. For traditional MoE models, many copies of the full-fledged feed-forward blocks need to be stored in memory at inference time which is an expensive undertaking. With our methods, only a single copy of the model backbone needs to be stored in memory in addition to lightweight parameter-efficient experts. This leads to a significant reduction in the memory requirements at inference time.

## 3 EXPERIMENTS

**Dataset** We conduct instruction-tuning experiments using a comprehensive set of prompt instructions from the Public Pool of Prompts (P3) dataset Sanh et al. (2022). We follow the same procedure as Raffel et al. (2020), where each task is converted into the format provided templates in (Sanh et al., 2022). P3 is a collection of 62 datasets covering a wide variety of tasks. For evaluation, we follow the same zero-shot evaluation presented in T0 (Sanh et al., 2022), which includes 8 datasets from different unseen task categories.

---

[2]The corresponding pseudo-code for MoLORA is given in Appendix A.9

**Experimental Setup** For the base pretrained models, we use T5 v1.1+LM adaptation (Lester et al., 2021) that includes T5 models of different sizes ranging from 770M to 11B parameters. For all experiments, we fine-tune using Adafactor optimizer (Shazeer & Stern, 2018) with a learning rate of $3e^{-4}$. We set the sequence length to 1024 for the input and 256 for the target following Sanh et al. (2022). For all parameter-efficient MoE variants, we fine-tune T5 models using a batch size of 32 over 500K steps.

**Baselines** We compare our mixture of parameter-efficient experts against both T0 baseline as the fully fine-tuned model and the standard parameter-efficient fine-tuning methods $(IA)^3$ and LORA with rank 4 for all model sizes. Additionally, for the 3B parameter base model, we evaluate LoRA with ranks 8 and 16. For T0 baselines, based on our experiments with different hyperparameters, we find that a larger batch size and learning rate result in better performance; thus, we replicated T0 by fine-tuning for 10k steps with a batch size of 256 and a learning rate of $1e^{-3}$, following Phang et al. (2023) – these hyperparameters achieve significantly higher results, as shown in Table 1. For $(IA)^3$ and LORA, we use the same training hyperparameters, such as a learning rate of $3e^{-4}$ and batch of 32 over 500k steps.

**Infrastructure** All experiments were conducted on TPU v4 machines up to 256 pod slices. We used SeqIO and T5X (Roberts et al., 2022) frameworks for training, evaluation, and inference of all the models experimented.

## 3.1 ABLATIONS

Given that no work to date has studied MoE in extremely parameter-efficient settings at scale, we also seek to understand the key characteristics of our proposed methodology by running rigorous ablations. We detail both briefly, along with the experimental setup below:

**Routing Input: Token vs Sentence Embeddings** *Do sentence embeddings in routing lead to higher performance?* In our main MoV and MoLORA methods, router layers take intermediate embeddings of input tokens as input, similar to other MoE architectures (Shazeer et al., 2017; Fedus et al., 2022). However, as an alternative, a sentence embedding can be computed for each instruction (prompt with corresponding input) and used as input for the router (Ye et al., 2022). To compare both, we use sentence embeddings that are derived using the Sentence-T5 (Ni et al., 2022), which was initialized from the pretrained T5 and trained on diverse data sources. Without additional fine-tuning, we pass each instruction to Sentence-T5 to obtain an embedding with a dimension of 768.

**Routing Strategy: Soft vs Discrete** *What is the best routing strategy in parameter-efficient MoEs?* In our MoE framework, we use soft merging of experts as the routing strategy. Soft merging refers to a weighted average of all the experts computed within a specified routing block. As an alternative, we also test a discrete top-k routing strategy as used in standard MoE architectures (Shazeer et al., 2018; Fedus et al., 2022). In the top-k routing approach, rather than considering all experts for a decision, only the top 'k' experts, determined by the router, are chosen for the computation which introduces the sparsity and decreases the amount of computation Note that, although the computation is conditional to the top-k experts, the required memory depends on the total number of experts.

We evaluate top-k selection with $k = \{1, 2\}$ as they were proposed by previous work (Shazeer et al., 2017; Fedus et al., 2022). Additionally, we assess discrete routing with top-k using *load balancing* following to Fedus et al. (2022) which promotes balanced top-k selection through an auxiliary loss, aiming for an equitable workload distribution among experts.

## 4 RESULTS AND DISCUSSION

**Parameter efficient MoEs vs PEFTs** *How does our MoE recipe compare to a single expert PEFT?* Table 1 compares zero-shot performance of PEFTs methods ($(IA)^3$ and LORA), and our variants of parameter-efficient MoE (MoV and MoLORA), using T5-3B as the base model. We observe that our MoE variants (MoV and MoLORA) present a significant performance boost over the standard $(IA)^3$ vectors and LORA adapters.

MoV using 30 experts achieves a 14.57% performance improvement compared to its dense counterpart $(IA)^3$. This improvement is consistent across all unseen tasks and is achieved at a marginal increase in

**Zero-shot Results at 3B Scale**

|  | Model | % Params. | ANLI | CB | RTE | WSC | WIC | Copa | WNG | HS | Average |
|---|---|---|---|---|---|---|---|---|---|---|---|
| *Full-FT* | T0-3B (Sanh et al., 2022) | 100% | 33.46 | 50.0 | 64.08 | 64.42 | 50.39 | 74.92 | 50.51 | 27.51 | 51.91 |
| | T0-3B (our replication) | 100% | 41.08 | 80.36 | 76.17 | 53.37 | 53.92 | 88.94 | 57.46 | 29.19 | 60.06 |
| *PEFT* | (IA)[3] | 0.018% | 34.08 | 50.0 | 66.43 | 56.25 | 55.41 | 79.08 | 52.09 | 29.91 | 52.90 |
| | LORA (rank 4) | 0.3% | 37.5 | 75.57 | 73.53 | 61.02 | 51.25 | 83.6 | 54.33 | 25.32 | 57.51 |
| | LORA (rank 8) | 0.6% | 37.5 | 75.0 | 77.98 | 62.5 | 51.49 | 83.67 | 55.72 | 27.3 | 58.89 |
| | LORA (rank 16) | 1.2% | 38.5 | 80.36 | 76.71 | 63.46 | 51.02 | 84.5 | 54.7 | 27.11 | 59.54 |
| *Our Method* | MoV-10 | 0.32% | 38.92 | 75.0 | 78.88 | 62.5 | 52.19 | 85.77 | 55.96 | 30.24 | 59.93 |
| | MoV-30 | 0.68% | 38.7 | 78.57 | 80.87 | 63.46 | 51.1 | 87.25 | 56.27 | 28.63 | 60.61 |
| | MoV-60 | 1.22% | 38.83 | 76.79 | 74.55 | 60.1 | 52.66 | 89.79 | 55.49 | 30.47 | 59.83 |
| | MoLORA-2 (rank 4) | 0.75% | 39.2 | 82.14 | 80.32 | 62.5 | 50.39 | 80.58 | 57.38 | 28.47 | 60.12 |
| | MoLORA-10 (rank 4) | 3.18% | 38.5 | 78.57 | 78.16 | 63.46 | 50.86 | 86.5 | 55.41 | 26.72 | 59.77 |
| | MoLORA-15 (rank 4) | 4.69% | 40.0 | 80.36 | 80.51 | 62.98 | 50.86 | 89.0 | 55.33 | 27.3 | 60.79 |

Table 1: Average median results on unseen tasks for full model fine-tuning (T0), parameter-efficient fine-tune methods ((IA)[3] and LORA) and our mixture of parameter-efficient experts (MoV and MoLORA), using T5-3B base model (Raffel et al., 2020). Note that our T0 scores are significantly higher than the original T0 confirming previous work (Phang et al., 2023; Ivison et al., 2023).

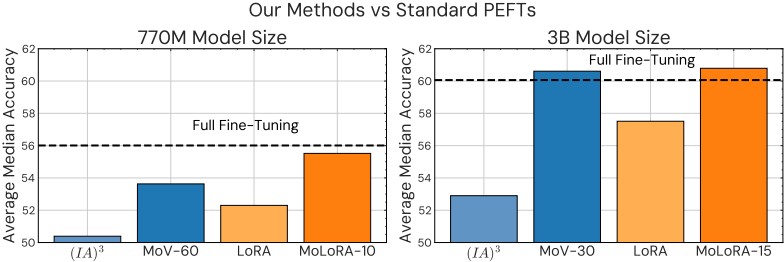

Figure 3: Comparison of the top-performing variants from our proposed mixture of PEFT experts versus their dense counterparts across T5-Large (*Left*) and T5-XL (*Right*).

the number of updated parameters – only an additional 0.018% parameters per expert. [3] In the context of LORA at rank 4, our MoLORA equipped with 15 experts, achieves an average median score increase of 5.70%. This improvement is notably less significant when compared to MoV. We attribute this disparity to the difference in updated parameter count in LORA adapters and (IA)[3] vectors (0.3% vs 0.018%). Finally, comparing LoRA rank 8 and 16 with MoV-30 covering similar parameter budgets, MoV-30 outperforms LoRA of both ranks. Overall, learning a mixture for both MoV and MoLORA as opposed to a single dense model leads to notable gains in zero-shot performance.

**MoV outperforms MoLORA given same parameter budget** Between our methods, MoV achieves a better performance-parameter cost trade-off at the 3B parameters base model. As shown in the left plot in figure 1 MoV with 30 experts, only updating 0.68% of all parameters, achieves nearly the same performance as MoLORA with 15 experts that updates 4.69% of parameters. This shows the effectiveness of our MoE approaches even with tiny experts at a large base model scale.

**Parameter efficient MoEs vs full fine-tuning** *How does MoE compare to updating all parameters during finetuning?* As shown in Table 1 when compared to fully fine-tuned T0-3B, our proposed methods, MoV and MoLORA both with 10 experts, are on par with full fine-tuning. This is impressive as MoV-10 only updates 0.32% of all model parameters. Furthermore, when increasing the number of experts from 10 to 15 and 30 for MoV and MoLORA respectively, our both methods outperform the full fine-tuning by a small margin.

### 4.1 HOW DO PARAMETER-EFFICIENT MoEs SCALE WITH BASE MODEL SIZE?

Figure 1 (right) shows the scaling characteristic of MoV with 60 experts compared with (IA)[3] and full fine-tuning for 770M, 3B and 11B parameters base models. We find that across all model sizes

---

[3]We provide comparison for fine-tuning efficiency between MoV and (IA)[3] in Appendix A.4.

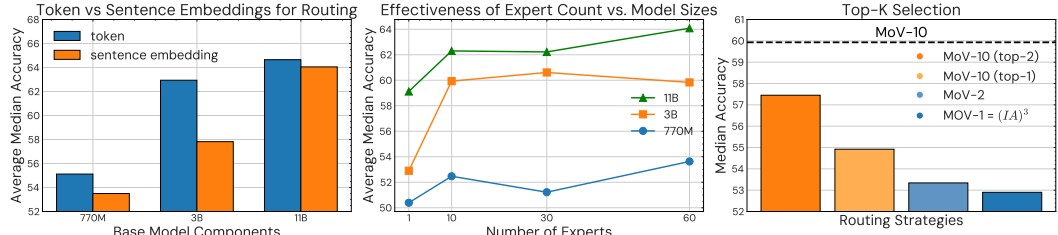

Figure 4: *Left:* Zero-shot performance of passing embedding of the token sequence to the router vs. passing tokens to the router. *Middle:* Zero-shot performance across T5 model sizes (Large, XL, XXL) as the number of experts increases. *Right:* The effectiveness of activating top-k experts.

we evaluate, our parameter-efficient MoEs consistently maintain higher performance compared to standard PEFTs and achieve comparable results with full fine-tuning.

**MoV benefits from scaling** At all model sizes, MoV-60 significantly outperforms standard $(IA)^3$. It is also far closer in performance to full fine-tuning than a single expert. For example, at 770M parameters, there is a 12.34% performance gap between $(IA)^3$ and full fine-tuning vs 5.56% for MoV-60. As the base model scales up, MoV becomes more competitive with full fine-tuning. For 3B and 11B parameter models, MoV-60 achieves performance approximately on par with the full fine-tuning, despite updating less than 1.3% of the total parameters.

**MoLORA outperforms MoV in smaller model size regimes** As discussed in the main results, at larger model sizes MoV achieves a better performance-parameter efficiency trade-off compared to MoLORA. Conversely, at the 770M scale, MoLORA with 10 experts that update 3.18% of total parameters, performs better compared to MoV-60 and nearly matches the performance of full fine-tuning (Figure 3). Finally, similar to MoV, MoLORA archives higher performance than LORA at both 770M and 3B scales.

## 4.2 How does the number of experts impact the downstream performance?

The center plot of Figure 4 shows the performance of MoV with different numbers of experts at all model sizes. We find that increasing the number of experts generally improves unseen task performance. However, this improvement is contingent upon the specific number of experts and the base model size. For both 770M and 11B parameter base models, our MoV method achieves its best performance by using 60 experts. To illustrate, when the number of experts is increased from 10 to 60, the average median accuracy improves from 52.47 to 53.63 for the 770M model and from 62.3 to 64.08 for the 11B model. However, the 3B model, using just 30 experts, updating 0.68% of the parameters, reaches peak accuracy with a score of 60.61 at this scale, as performance stagnates when 60 experts are used. This trend of performance improvement by scaling more experts is further endorsed in the context of MoLORA; when scaling experts from sets of (5, 10, 15), there was a corresponding elevation in the average median score, registering at 58.6, 59.77, and 60.79, respectively.

## 4.3 What is the best routing strategy in parameter-efficient MoEs?

In Figure 4, the rightmost plot shows the overall unseen task performance when using different routing strategies for MoV. Specifically, we compare the *soft merging* of 10 experts (dashed line) with discrete top-2 and top-1 routing. We observe that soft merging significantly outperforms discrete routing in the MoV-10 setting. Specifically, for discrete routing with top-k experts, where k is 1 and 2, the MoE achieves an average median accuracy of 54.92 and 57.45 respectively. In contrast, using the soft merging approach, where all experts are activated, we observe an accuracy of 59.93.

Furthermore, to understand if we recover the performance loss of top-k routing by using load balancing, we integrated the loss balancing following to Fedus et al. (2022). However, we find that the top-k selection of $k = 2$ with load balancing loss leads to a further decrease in performance 1.5 average median score. Together, these results show that in extremely parameter-efficient MoE settings, soft merging enables superior performance. Note that top-2 and top-1 routing strategies (among 10

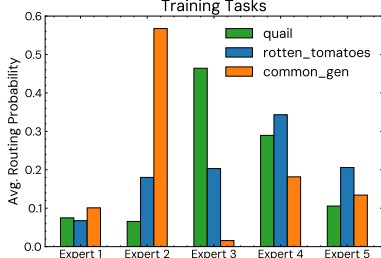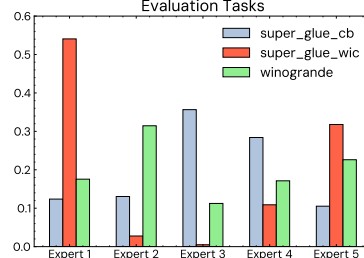

Figure 5: Mean expert routing probabilities for intermediates activations at the last feedforward layer. Values are averaged across tokens and batch. Experts are weighted differently in soft merging depending on the task. *Left:* Measured on tasks seen during training. *Right:* Measured on unseen evaluation tasks.

experts) perform better than MoV with only 2 experts and a single expert (IA)[3] respectively, showing that soft merging performs better when a larger number of experts are used.

### 4.4 DOES SENTENCE EMBEDDINGS IN ROUTING LEAD TO HIGHER PERFORMANCE?

To understand the effects of a pronounced inductive bias towards task representations in our MoE framework, we compare using sentence embeddings of instructions with token embeddings for the routing input. These sentence embeddings are obtained offline using an external sentence embedding model. Here, we aim to evaluate how pronounced task information affects the router's decision and the subsequent generalization capabilities of the model in downstream tasks. Figure 4 leftmost plot shows performances of token routing and sentence routing at all model sizes. We find that the token routing exhibits superior performance with 3.03%, 8.86%, and 0.94% improvement for 770M, 3B, and 11B base model sizes, respectively. These results suggest that a higher degree of inductive bias for task datasets is not necessarily beneficial, as our approaches can acquire a diverse set of task knowledge directly from the hidden representations of tokens. Furthermore, token routing enables the use of learned experts and routing layers without any prior task information for unseen tasks.

### 4.5 DO EXPERTS SPECIALIZE IN DIVERSE KNOWLEDGE ACROSS DIFFERENT TASKS?

To understand how expert routing differs for different tasks, we take a closer look at how experts are activated for a variety of tasks. Figure 5 shows the mean expert probabilities for MoV with 5 experts that are located in feed-forward layers in the last decoder block at the 770M parameter T5 model. We selected the last decoder block as it has been shown that deeper layers learn more task-specific information (Rogers et al., 2020). We plot the mean routing probabilities for both training tasks and evaluation tasks that are unseen during training to understand cross-task generalization through the lens of experts if skills learned at training time generalize to unseen tasks at evaluation time. Intuitively, if experts have indeed learned different *skills*, we expect that they contribute in different degrees to tasks that are different in nature. The amount of contribution is directly reflected in the routing probability of each expert since we use soft merging, i.e., summation of expert vectors weighted by the routing probability as described in Figure 2. As such, the mean routing probabilities plotted in Figure 5 provide an overall picture of the contribution of each expert, depending on the downstream task.

**Specialization across unseen vs seen tasks** As depicted in Figure 5, both evaluation and training tasks lead to the activation of experts at different magnitudes. For example, both `quail` and `super_glue_cb` activate Expert 3 the most out of the 5 experts, followed by Expert 4, but are different both in terms of the relative contribution of each expert and the ordering of the remaining 3 experts based on routing probability. A similar pattern can be observed for `common_gen` & `winogrande`, as they both activate Expert 2 the most but are otherwise different. Overall, the fact that routing specialization seems to occur *regardless* of whether the downstream task was trained on, suggests that expert specialization is inherent and transferable from seen tasks to unseen tasks.

## 4.6 HYPERPARAMETERS SENSITIVITY

Given the widely documented sensitivity of MoE-style architecture to hyperparameters (Fedus et al., 2022; Shazeer et al., 2017), we ran extensive ablation studies to uncover the idiosyncrasy of PEFT methods in the context of MoE. We experimented with batch sizes of 32, 128, 256, and 2048 and we found that the larger the batch size, the more likely our MoEs to collapse to a single expert. Our empirical finding resonates with Shen et al. (2023) which also finds that a small batch is necessary for stable training. For instance, by experimenting with a batch size of 2048 and evaluating every 5K steps up to 20K, we observed that the performance of our parameter-efficient MoEs deteriorates after 5K steps, converging to performance levels akin to their dense counterparts. Additionally, we experimented with varying learning rates from $3e^{-3}$ to $6e^{-4}$. We find that a smaller learning rate $3e^{-4}$ stabilizes training in parameter-efficient experts by preventing rapid, imbalanced updates that can suppress diversity and lead to suboptimal solutions.

## 5 RELATED WORK

**Mixture-of-Experts (MoE)** MoE has been deeply explored in Natural Language Processing (Shazeer et al., 2017; Lepikhin et al., 2020; Kudugunta et al., 2021; Lou et al., 2022; Mustafa et al., 2022; Fedus et al., 2022; Du et al., 2022; Zoph et al., 2022; Clark et al., 2022; Zhou et al., 2022; Zuo et al., 2022; Komatsuzaki et al., 2023). It effectively increases the model's parameter capacity, activating specific parts while maintaining similar computation to its dense equivalent. Notably, efforts have been made to improve routing (Eigen et al., 2014; Hazimeh et al., 2021; Lewis et al., 2021; Roller et al., 2021; Zhou et al., 2022; Zuo et al., 2022; Fedus et al., 2022; Du et al., 2022). MoE's utility also has been extended to multi-task scenarios (Hazimeh et al., 2021; Kudugunta et al., 2021). Unlike these, our research scales both data volume and task count in MoE, targeting its training instability while prioritizing efficient fine-tuning. Recently, Shen et al. (2023) discussed instruction fine-tuning to address the generalization challenges tied to MoE models. We explore its efficacy, emphasizing a unique PEFT component to counter the high memory demands of traditional MoE.

**Parameter-Efficient Fine-tuning**. Houlsby et al. (2019) established "adapters" in the NLP domain to fine-tune BERT. Many adapter variants exist with varied design choices (Bapna et al., 2019; Pfeiffer et al., 2021; Li & Liang, 2021; Hu et al., 2021; Zaken et al., 2022; Liu et al., 2022). We have chosen (IA)$^3$ (Liu et al., 2022) and LORA (Hu et al., 2021) as PEFT components in our MoE because they offer an optimal balance between performance and parameter efficiency (Mahabadi et al., 2021).

Several studies have explored PEFT in relation to MoE, with nuances. Wang et al. (2022) studied single-task fine-tuning by employing random routing and few-shot evaluation. In contrast, our research focuses on instruction-tuning across multiple tasks, using up to an 11B parameter T5 model Raffel et al. (2020). We emphasize its potential by evaluating unseen tasks through instruction tuning. In Ponti et al. (2022), they learn 'skills' and a binary matrix to determine tasks and skills relationship. Building on this, Caccia et al. (2023) decomposes task-skill matrix introducing a 'multi-head' routing function, exploring supplementary inductive biases, and showing effectiveness. Unlike the two mentioned studies, we fine-tunes without specific task identifiers, enabling evaluation of unseen tasks without further few-shot fine-tuning. Contemporaneously, Maxine (2023) uses a routing method based on cosine distances between the embedding of input sentences and the average embedding of examples in each dataset used to train experts separately, unlike our end-to-end training of all experts.

## 6 CONCLUSION

This work introduces parameter-efficient MoEs, Mixture of Vectors (MoV) and Mixture of LoRAs (MoLORA), to address the challenges of standard MoEs in a strictly computationally limited environment. Our method outperforms parameter-efficient techniques and achieves performance parity with full fine-tuning on unseen tasks by updating less than 1% of the 3B and 11B model parameters. Our extensive experiments, including rigorous ablations across model sizes, routing input and routing strategy, confirm the effectiveness of our approach across diverse unseen tasks, highlighting its superior accuracy and computational efficiency. Furthermore, our framework's versatility seamlessly integrates with other parameter-efficient strategies and remains compatible with efficiency-enhancing techniques such as quantization (Dettmers et al., 2023).

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

# A   APPENDIX

## A.1   EXTENDED RELATED WORK

**Mixture-of-Experts**   The Mixture-of-Experts (MoE) has been investigated thoroughly in Natural Language Processing (Lou et al., 2022; Mustafa et al., 2022; Shazeer et al., 2017; Lepikhin et al., 2020; Fedus et al., 2022; Du et al., 2022; Zoph et al., 2022; Clark et al., 2022; Zhou et al., 2022; Komatsuzaki et al., 2023; Kudugunta et al., 2021; Zuo et al., 2022) as an effective way of increasing the model's capacity in parameter size where certain parts of the model are activated while computation is kept the same or close to its dense counterpart. In the context of MoE, there is a body of work focusing on improving the routing (Hazimeh et al., 2021; Lewis et al., 2021; Roller et al., 2021; Zhou et al., 2022) including random routing (Zuo et al., 2022) activating all expert through weighted average (Eigen et al., 2014) to sparsely select a single or $k$ experts (Fedus et al., 2022; Du et al., 2022). MoE has also been invested in multi-task settings including multilingual neural machine translation(Hazimeh et al., 2021; Kudugunta et al., 2021). Unlike these studies, our research addresses MoE by scaling both the volume of data and the number of tasks, aiming to mitigate the instability inherent in training the MoE models. But our primary emphasis remains on achieving efficient fine-tuning. Recently, Shen et al. (2023) highlighted how instruction fine-tuning with scaled tasks can counteract the generalization challenges tied to MoE models. In distinction from this, our study scrutinizes the efficacy of instruction fine-tuning in the MoE domain, specifically concentrating on a unique ensemble of the PEFT components, considering the memory cost of the traditional MoE can be prohibitive for many practitioners. Similar to the aforementioned work, Ye et al. (2022) utilized MoE in a multi-task context, employing BART Lewis et al. (2019) as their pre-trained model. However, they limited their experimental scope to a smaller scale and used replicas of each transformer layer as experts, simply multiplying the model by the number of experts. Our work, on the other hand, presents an extreme parameter efficiency with small experts at a large scale up to 11B parameter base model.

**Instruction Tuning**   Instruction tuning, as elucidated in (Sanh et al., 2022; Wei et al., 2022; Mishra et al., 2022), is a technique where a language model is fine-tuned over a collection of tasks using paired prompts and responses. The primary goal of this technique is to enable the model to predict responses accurately based on the provided prompts, thereby augmenting its ability to understand and execute instructions effectively. The method has gained considerable attention due to its pronounced success in enhancing zero-shot performance on tasks to which the model has not been previously exposed. Additionally, instruction tuning has led to breakthroughs such as Chain of Thought Prompting (Wei et al., 2023) where a breakdown of complex problems into smaller steps to produce intermediate reasoning along with the final solution, PaLM (Chowdhery et al., 2022), FLAN (Wei et al., 2022). In our work, we explore the use of instruction fine-tuning with the intention of harnessing its benefits that enable the model to learn from a diverse set of inputs where the mixture of expert style models

suits well, for enhanced evaluation performance on unseen tasks. Our objective remains to optimize computational efficiency without compromising zero-shot performance.

**Parameter-Efficient Fine-tuning**. Houlsby et al. (2019) established "adapters" in the NLP domain to fine-tune BERT. There are many variants of adapters with different design choices (Bapna et al., 2019; Pfeiffer et al., 2021). Li & Liang (2021) proposed updating soft prompts concatenated to embeddings or layer outputs instead of adapters. Zaken et al. (2022) show that just updating only a small subset of parameters during fine-tuning (e.g. just biases) is very effective. Hu et al. (2021) proposed LORA based on low-rank decomposition matrices of transformer layers. They show superior performance with a smaller parameter budget and no inference cost as LORA parameters can be applied offline to the baseline model. Liu et al. (2022) proposed $(IA)^3$, task-specific vectors to modify attention activation. Instead of using feedforward layers inserted in transformer layers as adapters, they learn vectors to update (by broadcast multiplication) key, value, and linear layer weight matrices. Unlike the other PEFT methods, $(IA)^3$ does not induce any additional inference cost and enables mix-batches (from different datasets). The multiplicative nature of the $(IA)^3$ creates an interesting opportunity for the mixture-of-expert type of modeling without parallelization overhead. Chen et al. (2023) experiment with different design spaces (essentially a hyperparameter search) for PEFT. They suggest four phases: 1) grouping layers into different sets; 2) adding trainable parameters towards each group; 3) deciding which group should be trained; 4) assigning groups with different training strategies. Their finding is that different architectures have different best settings. We have chosen $(IA)^3$ and LORA as our PEFT components because they offer an optimal balance between performance and parameter efficiency (Mahabadi et al., 2021; Liu et al., 2022).

Several studies have explored PEFT in the context of MoE or in a similar fashion, albeit with certain distinctions. For instance, Wang et al. (2022) focused on single-task fine-tuning employing a mixture of adapters for $BERT_{base}$ with 110M parameters (Devlin et al., 2019) and $RoBERTa_{large}$ with 355M parameters (Liu et al., 2019), incorporating random routing, and adopting a few-shot evaluation. In divergence from this, our work centers on instruction-tuning with multiple tasks present during fine-tuning. We underscore the efficacy of this approach by rigorously testing up to 11B parameter text-to-text model Raffel et al. (2020), implementing token routing, and strictly emphasizing evaluation on a set of unseen (held-out) tasks to underscore the potential of instruction tuning. In another work, Ponti et al. (2022) introduced Polytropon, which involves learning adapters (termed as 'skills') specific to each task and employing a task-skills binary matrix to determine the skill set associated with each task. In their method, input examples dictate the selection of adapters. These adapters are then aggregated, and the resultant single adapter is integrated into the overall architecture. Extending upon the Polytropon framework, Caccia et al. (2023) implemented a distinct skill set for every layer in their variant named Polytropon-S. They introduce a deterministic routing function, delve into supplementary inductive biases, show effectiveness up to 3B models, and they don't employ MoE style architecture. Our research presents a departure from these two studies. Specifically, our primary experimental setup employs MoEs that do not require any specific task identifier during fine-tuning by the use of the token routing strategy. In this way, we can evaluate our instruction-tuned MoEs on unseen tasks without any further task-specific few-shot fine-tuning. Contemporaneously, Maxine (2023) uses a routing method based on cosine distances between the embedding of input sentences and the average embedding of examples in each dataset used to train experts separately, unlike our end-to-end training of all experts. We showed the scaling property of our MoEs in this setting by fine-tuning models up to 11B parameters.

## A.2    ZERO-SHOT EVALUATION FOR P3 DATASET

In our study, we conducted a comprehensive evaluation of the variants of our proposed methods in comparison to our established baselines. This evaluation encompassed various sizes of the T5 model, specifically 770M, 3B, and 11B. These results are given in Table 2, 3 and 4. Both mean and median scores were reported for every evaluation set derived from the P3 dataset, which covers a range of tasks. For further details and a more in-depth exploration, please refer to the following URL: `https://huggingface.co/datasets/bigscience/P3`.

**T5-Large (770M)**

| | Model | % Params. | Metric | ANLI | CB | RTE | WSC | WIC | Copa | WNG | HS | Average |
|---|---|---|---|---|---|---|---|---|---|---|---|---|
| *Full-FT* | T0-770M (ours) | 100% | median | 35.6 | 71.43 | 75.63 | 57.21 | 51.41 | 77.0 | 53.04 | 26.78 | 56.01 |
| | | | mean | 35.57 | 57.74 | 75.88 | 52.31 | 52.52 | 74.6 | 52.93 | 26.74 | 53.54 |
| *PEFT* | (IA)$^3$ | 0.036% | median | 33.5 | 42.86 | 67.87 | 62.02 | 52.35 | 67.0 | 51.22 | 26.33 | 50.39 |
| | | | mean | 33.27 | 45.12 | 67.08 | 58.17 | 52.74 | 66.63 | 51.35 | 26.32 | 50.09 |
| | LoRA (rank 4) | 0.497% | median | 35.0 | 55.36 | 57.4 | 63.46 | 50.24 | 77.0 | 53.28 | 26.67 | 52.3 |
| | | | mean | 35.26 | 51.67 | 59.35 | 62.98 | 50.66 | 76.5 | 52.41 | 27.24 | 52.0 |
| | LoRA (rank 40) | 5% | median | 35.75 | 64.29 | 58.84 | 63.46 | 50.31 | 78.50 | 52.94 | 26.81 | 53.86 |
| | | | mean | 35.54 | 51.55 | 59.68 | 63.56 | 50.63 | 76.61 | 52.47 | 26.91 | 52.13 |
| *Our Method* | MOV-5 | 0.27% | median | 33.6 | 41.07 | 71.48 | 61.54 | 50.86 | 76.5 | 51.46 | 26.02 | 51.57 |
| | | | mean | 33.51 | 42.62 | 71.26 | 60.96 | 51.14 | 73.8 | 51.55 | 26.01 | 51.36 |
| | MoV-10 | 0.55% | median | 33.9 | 42.86 | 74.19 | 62.5 | 50.31 | 77.0 | 52.64 | 26.34 | 52.47 |
| | | | mean | 33.68 | 42.38 | 74.51 | 59.23 | 50.74 | 74.82 | 52.2 | 26.72 | 51.78 |
| | MoV-20 | 1.10% | median | 33.7 | 41.07 | 73.83 | 63.46 | 50.94 | 75.46 | 51.14 | 25.48 | 51.89 |
| | | | mean | 33.98 | 45.12 | 73.36 | 59.13 | 51.33 | 73.47 | 51.3 | 25.45 | 51.64 |
| | MoV-30 | 1.66% | median | 33.75 | 41.07 | 72.92 | 55.77 | 51.25 | 77.0 | 51.46 | 26.55 | 51.22 |
| | | | mean | 33.81 | 44.88 | 72.56 | 56.15 | 51.29 | 77.43 | 51.81 | 26.52 | 51.81 |
| | MoV-60 | 3.32% | median | 34.0 | 53.57 | 75.81 | 57.69 | 50.55 | 77.96 | 53.12 | 26.33 | 53.63 |
| | | | mean | 34.24 | 52.26 | 75.02 | 58.37 | 50.78 | 77.06 | 52.87 | 26.74 | 53.42 |
| | MoLoRA-10 | 5.60% | median | 33.2 | 67.86 | 68.41 | 64.9 | 50.39 | 80.0 | 52.64 | 52.64 | 55.52 |
| | | | mean | 33.37 | 56.31 | 68.88 | 63.37 | 51.55 | 79.35 | 52.31 | 52.31 | 53.99 |

Table 2: Zero-shot evaluation of the 770M parameter model across all unseen tasks, comparing different numbers of experts for both MoV and MoLoRA.

**T5-XL (3B)**

| | Model | % Params. | Metric | ANLI | CB | RTE | WSC | WIC | Copa | WNG | HS | Average |
|---|---|---|---|---|---|---|---|---|---|---|---|---|
| *Full-FT* | T0-3B (Sanh et al., 2022) | 100% | median | 33.46 | 50.0 | 64.08 | 64.42 | 50.39 | 74.92 | 50.51 | 27.51 | 51.91 |
| | | | mean | 33.42 | 45.36 | 64.55 | 65.10 | 50.69 | 72.40 | 50.97 | 27.29 | 51.22 |
| | T0-3B (our replication) | 100% | median | 41.08 | 80.36 | 76.17 | 53.37 | 53.92 | 88.94 | 57.46 | 29.19 | 60.06 |
| | | | mean | 40.73 | 74.52 | 76.82 | 52.21 | 53.84 | 88.99 | 56.83 | 29.2 | 59.14 |
| *PEFT* | (IA)$^3$ | 0.018% | median | 34.08 | 50.0 | 66.43 | 56.25 | 55.41 | 79.08 | 52.09 | 29.91 | 52.90 |
| | | | mean | 34.56 | 51.07 | 68.38 | 54.9 | 55.61 | 78.23 | 52.14 | 28.97 | 52.98 |
| | LoRA (rank 4) | 0.3% | median | 37.5 | 75.57 | 73.53 | 61.02 | 51.25 | 83.6 | 54.33 | 25.32 | 57.51 |
| | | | mean | 37.85 | 66.9 | 77.04 | 56.73 | 52.29 | 82.83 | 55.64 | 26.79 | 57.01 |
| | LoRA (rank 8) | 0.6% | median | 37.5 | 75.0 | 77.98 | 62.5 | 51.49 | 83.67 | 55.72 | 27.3 | 58.89 |
| | | | mean | 37.64 | 64.05 | 77.91 | 56.71 | 51.77 | 82.84 | 55.23 | 26.83 | 57.03 |
| | LoRA (rank 16) | 1.2% | median | 38.5 | 80.36 | 76.71 | 63.46 | 51.02 | 84.5 | 54.7 | 27.11 | 59.54 |
| | | | mean | 37.11 | 65.6 | 77.62 | 60.48 | 51.49 | 82.29 | 55.14 | 26.71 | 57.05 |
| *Our Method* | MoV-2 | 0.18% | median | 34.7 | 46.43 | 66.06 | 56.25 | 54.86 | 85.42 | 53.75 | 29.25 | 53.34 |
| | | | mean | 35.14 | 50.36 | 69.31 | 56.15 | 54.4 | 83.79 | 53.69 | 28.47 | 53.91 |
| | MoV-5 | 0.23% | median | 37.1 | 76.79 | 78.16 | 57.69 | 52.27 | 86.77 | 53.99 | 29.31 | 59.01 |
| | | | mean | 37.66 | 62.14 | 78.3 | 58.46 | 53.54 | 86.52 | 54.54 | 28.3 | 57.43 |
| | MoV-10 | 0.32% | median | 38.92 | 75.0 | 78.88 | 62.5 | 52.19 | 85.77 | 55.96 | 30.24 | 59.93 |
| | | | mean | 38.83 | 63.45 | 79.49 | 60.19 | 53.04 | 86.41 | 56.27 | 29.11 | 58.35 |
| | MoV-20 | 0.50% | median | 39.2 | 75.0 | 76.71 | 57.69 | 53.45 | 89.0 | 55.64 | 30.89 | 59.7 |
| | | | mean | 39.25 | 64.05 | 76.53 | 56.63 | 53.45 | 86.93 | 56.24 | 29.36 | 57.81 |
| | MoV-30 | 0.68% | median | 38.7 | 78.57 | 80.87 | 63.46 | 51.1 | 87.25 | 56.27 | 28.63 | 60.61 |
| | | | mean | 38.9 | 67.5 | 81.23 | 59.9 | 52.43 | 86.28 | 56.39 | 27.57 | 58.77 |
| | MoV-60 | 1.22% | median | 38.83 | 76.79 | 74.55 | 60.1 | 52.66 | 89.79 | 55.49 | 30.47 | 59.83 |
| | | | mean | 38.97 | 63.93 | 75.38 | 57.79 | 53.5 | 86.04 | 55.88 | 29.28 | 57.59 |
| | MoV-10 (top-1) | 0.32% | median | 33.9 | 75.0 | 71.12 | 61.06 | 50.71 | 70.0 | 51.7 | 25.89 | 54.92 |
| | | | mean | 34.31 | 60.6 | 71.41 | 58.94 | 51.24 | 68.39 | 51.79 | 25.98 | 52.82 |
| | MoV-10 (top-2) | 0.32% | median | 38.7 | 82.14 | 75.63 | 48.08 | 53.68 | 79.88 | 54.14 | 27.37 | 57.45 |
| | | | mean | 38.89 | 69.76 | 74.95 | 47.69 | 53.51 | 79.89 | 53.83 | 26.91 | 55.67 |
| | MoLORA-2 (rank 4) | 0.75% | median | 39.2 | 82.14 | 80.32 | 62.5 | 50.39 | 80.58 | 57.38 | 28.47 | 60.12 |
| | | | mean | 38.86 | 65.71 | 80.0 | 60.0 | 50.8 | 82.17 | 56.51 | 28.03 | 57.76 |
| | MoLORA-5 (rank 4) | 1.66% | median | 36.75 | 71.43 | 79.96 | 56.25 | 55.17 | 85.81 | 55.8 | 27.63 | 58.6 |
| | | | mean | 37.52 | 62.14 | 80.22 | 52.6 | 55.34 | 84.05 | 56.04 | 26.62 | 56.82 |
| | MoLORA-10 (rank 4) | 3.18% | median | 38.5 | 78.57 | 78.16 | 63.46 | 50.86 | 86.5 | 55.41 | 26.72 | 59.77 |
| | | | mean | 38.49 | 66.43 | 77.44 | 59.9 | 51.63 | 84.96 | 56.1 | 26.7 | 57.71 |
| | MoLORA-15 (rank 4) | 4.69% | median | 40.0 | 80.36 | 80.51 | 62.98 | 50.86 | 89.0 | 55.33 | 27.3 | 60.79 |
| | | | mean | 39.73 | 69.52 | 80.97 | 60.67 | 51.54 | 86.5 | 55.03 | 27.25 | 58.9 |

Table 3: In our most comprehensive experimental setup, we conducted a zero-shot evaluation across all unseen tasks using a 3B parameter model. We compared varying numbers of experts for both MoV and MoLoRA and experimented with a top-k selection routing strategy

## A.3 Token vs. Sentence Embeddings for Routing

We present the mean and median results for our routing strategies in Table 5. Specifically, we assessed performance by either passing tokens directly to the router or by passing sentence embeddings.

**T5-XXL (11B)**

| | Model | % Params. | Metric | ANLI | CB | RTE | WSC | WIC | Copa | WNG | HS | Average |
|---|---|---|---|---|---|---|---|---|---|---|---|---|---|
| Full-FT | T0-11B (Sanh et al., 2022) | 100% | median | 42.17 | 78.57 | 81.23 | 64.42 | 57.21 | 90.79 | 60.46 | 33.65 | 63.56 |
| | | | mean | 41.16 | 70.12 | 80.83 | 61.45 | 56.58 | 90.02 | 59.94 | 33.58 | 61.70 |
| | T0-11B (our replication) | 100% | median | 47.1 | 80.36 | 81.41 | 60.1 | 56.27 | 96.08 | 67.32 | 31.61 | 65.03 |
| | | | mean | 45.83 | 72.62 | 81.52 | 58.17 | 56.66 | 96.0 | 66.77 | 30.95 | 63.57 |
| PEFT | (IA)$^3$ | 0.0098% | median | 42.3 | 73.21 | 75.99 | 58.65 | 52.04 | 86.27 | 54.3 | 30.27 | 59.12 |
| | | | mean | 42.1 | 63.27 | 75.31 | 55.49 | 52.27 | 85.74 | 55.06 | 30.09 | 57.41 |
| Our Method | MoV-10 | 0.143% | median | 45.83 | 76.79 | 78.52 | 53.85 | 51.88 | 94.23 | 63.77 | 33.5 | 62.3 |
| | | | mean | 44.73 | 70.12 | 78.88 | 54.23 | 53.26 | 93.64 | 63.57 | 33.59 | 61.5 |
| | MoV-20 | 0.287% | median | 44.58 | 76.79 | 73.83 | 55.77 | 52.98 | 95.0 | 62.27 | 32.92 | 61.77 |
| | | | mean | 43.54 | 69.17 | 74.4 | 52.88 | 54.5 | 93.93 | 62.95 | 32.85 | 60.53 |
| | MoV-30 | 0.431% | median | 43.6 | 76.79 | 77.62 | 56.73 | 53.84 | 93.62 | 64.25 | 31.34 | 62.22 |
| | | | mean | 43.32 | 69.29 | 77.22 | 53.56 | 56.03 | 93.65 | 63.52 | 31.32 | 60.99 |
| | MoV-60 | 0.862% | median | 45.17 | 75.0 | 83.03 | 60.1 | 53.68 | 95.42 | 65.82 | 34.38 | 64.08 |
| | | | mean | 43.9 | 69.88 | 83.07 | 56.54 | 54.51 | 94.01 | 64.56 | 34.17 | 62.58 |

Table 4: We evaluated the largest available model size from the original T5 pre-trained checkpoint, T5-XXL with 11B parameters, to demonstrate the efficacy of our proposed mixture of PEFT experts at this scale.

Our findings indicate that, particularly for the T5-XL (3B) model, token routing consistently yields better performance in terms of both mean and median values. The Anli dataset is excluded from our embedding dataset.

**MoV – Token vs. Sentence Embedding**

| Model | Metric | CB | RTE | WSC | WIC | Copa | WNG | HS | Average |
|---|---|---|---|---|---|---|---|---|---|
| MoV-10 (Token) - 770M | median | 42.86 | 74.19 | 62.5 | 52.64 | 52.64 | 77.0 | 26.34 | 55.12 |
| | mean | 42.38 | 74.51 | 59.23 | 52.2 | 52.2 | 74.82 | 26.72 | 54.37 |
| MoV-10 (Embedding) - 770M | median | 48.21 | 67.15 | 62.98 | 51.8 | 50.99 | 67.0 | 26.38 | 53.5 |
| | mean | 51.67 | 67.29 | 58.37 | 51.79 | 50.99 | 65.8 | 26.57 | 53.21 |
| MoV-10 (Token) - 3B | median | 75.0 | 78.8 | 62.5 | 52.19 | 55.96 | 85.77 | 30.24 | 62.94 |
| | mean | 63.45 | 79.49 | 60.19 | 53.04 | 56.27 | 86.41 | 29.11 | 61.14 |
| MoV-10 (Embedding) - 3B | median | 57.14 | 67.15 | 61.06 | 55.33 | 52.49 | 82.5 | 29.08 | 57.82 |
| | mean | 51.07 | 68.81 | 58.65 | 55.28 | 52.57 | 80.53 | 28.51 | 56.49 |
| MoV-10 (Token) - 11B | median | 76.79 | 78.52 | 53.85 | 51.88 | 63.77 | 94.23 | 33.5 | 64.65 |
| | mean | 70.12 | 78.88 | 54.23 | 53.26 | 63.57 | 93.64 | 33.59 | 63.9 |
| MoV-10 (Embedding) - 11B | median | 75.0 | 78.7 | 57.69 | 54.0 | 57.85 | 92.0 | 33.08 | 64.05 |
| | mean | 66.19 | 79.1 | 58.37 | 54.83 | 58.78 | 91.17 | 32.7 | 63.02 |

Table 5: The above results demonstrate the effectiveness of token routing in comparison to imposing a strong inductive bias, such as sentence embedding across various model parameters.

### A.4 FINE-TUNING AND INFERENCE EFFICIENCY

We have curated a comprehensive table detailing the efficiency metrics, which include parameter number, training and inference time, memory consumption, and overall performance for various methods. By using these metrics, we aim to provide a more nuanced understanding of the performance-to-cost ratio.

Table 6 shows the updated parameter count and training time ratios for Mixture of Vectors (MoV) and (IA)$^3$ with respect to full-fine tuning. $(IA)^3$ trains fastest but with the worst performing method. MoV (10 and 30 experts) offers a balance of a large reduction in training speed and higher performance, outperforming LoRA in training time (1.3x) and overall zero-shot performance.

Table 7 shows the inference time metrics for T5-3B. In inference, $(IA)^3$ and LoRA add no extra time when their weights are integrated into the base model, while MoV increases it by only 1.06x to 1.15x depending on the sequence length and the number of experts. Furthermore, during the inference time, we experimented using only the top-K experts that are determined by the router probabilities to decrease the inference cost. We found that when a large number of experts are trained (e.g. 30), using only a top-K (e.g. 15) subset of these experts during inference maintains high performance. Concretely our MoV-30 with top-15 experts achieves an average performance of 60.25. This is very

| Base model size | Model | % Params. updated ↓ | % Training time reduction ↑ | Average zero-shot performance ↑ |
|---|---|---|---|---|
| *3B* | (IA)[3] | 0.018% | 38% | 52.90 |
| | MoV (10 Experts) | 0.32% | 31% | 59.93 |
| | MoV (30 Experts) | 0.68% | 27% | 60.61 |
| | LoRA (rank4) | 0.3% | 7% | 57.51 |
| | LoRA (rank 8) | 0.6% | 4% | 58.89 |
| | LoRA (rank 16) | 1.2% | 3% | 59.54 |
| *11B* | (IA)[3] | 0.0098% | 36% | 59.12 |
| | MoV (10 Experts) | 0.143% | 27% | 62.30 |
| | MoV (30 Experts) | 0.431% | 23% | 62.22 |

Table 6: Fine-tuning efficiency metrics for Mixture of Vectors (MoV) and its dense PEFT counterpart, (IA)[3]. We compare the ratio of updated parameters and the training times with respect to full fine-tuning (with the same batch size), across different scales. MoV exhibits a marginal difference in training time compared to (IA)[3] while demonstrating significant improvement in performance

competitive with using all experts (60.61), however, decreases the inference latency by 1% (1.14x) for the sequence length of 128.

Finally, Table 8 shows the peak memory usage during fine-tuning for MoV and the baselines using T5-3B as the base model. In terms of updated parameters vs performance trade-off, our MoV method outperforms others, while balancing parameter count and performance effectively. For the maximum memory use since we use *float32* computation for the router, MoV uses slightly more peak memory, however, it offers optimal performance without significant efficiency costs during training.

| Method | (IA)[3] | LoRA | *Seq length: 1024* MoV-10 | MoV-30 |
|---|---|---|---|---|
| Increase in latency | 1x | 1x | 1.06x | 1.07x |
| Method | (IA)[3] | LoRA | *Seq length: 512* MoV-10 | MoV-30 |
| Increase in latency | 1x | 1x | 1.08x | 1.11x |
| Method | (IA)[3] | LoRA | *Seq length: 128* MoV-10 | MoV-30 |
| Increase in latency | 1x | 1x | 1.10x | 1.15x |

Table 7: Relative increase in increase time using different sequence lengths for fine-tuning T5-3B model, measured in TPU v4-32 with per-replica batch size of 8)

| Method | (IA)[3] | LoRA (r=4) | LoRA (r=8) | LoRA (r=16) | MoV-10 | MoV-30 |
|---|---|---|---|---|---|---|
| Peak Mem (GB) | 7.88 | 8.96 | 9.01 | 9.05 | 10.44 | 10.62 |

Table 8: Peak memory usage for fine-tuning T5-3B model, measured in TPU v4-32

## A.5 MoLoRA vs. LoRA

It is important to clarify that multiple parallel LoRA in our MOE-style architecture differs in terms of optimization from single higher-rank LoRA due to the use of routers. By conditioning via the router that is sensitive to different input data, small LoRA weights in the MoLoRA can exploit different low-rank properties. Therefore, MoLoRA trained with MoE-style routing enables better "specialization" for different tasks in each smaller LoRA.

Unlike MoLoRA, a single higher-rank LoRA that lacks such router conditioning exploits general properties already emphasized in the base model. This limits specialization and can be suboptimal, especially when fine-tuning data is highly heterogeneous.

**MoLoRA vs. LoRA**

| Model | Metric | ANLI | CB | RTE | WSC | WIC | Copa | WNG | HS | Average |
|---|---|---|---|---|---|---|---|---|---|---|
| LoRA (rank 40) - 770M | median | 35.75 | 64.29 | 58.84 | 63.46 | 50.31 | 78.50 | 52.94 | 26.81 | 53.86 |
| | mean | 35.54 | 51.55 | 59.68 | 63.56 | 50.63 | 76.61 | 52.47 | 26.91 | 52.13 |
| MoLORA-10 (rank 4) - 770M | median | 33.2 | 67.86 | 68.41 | 64.9 | 50.39 | 80.0 | 52.64 | 52.64 | 55.52 |
| | mean | 33.37 | 56.31 | 68.88 | 63.37 | 51.55 | 79.35 | 52.31 | 52.31 | 53.99 |
| LoRA (rank 8) - 3B | median | 37.5 | 75.0 | 77.98 | 62.5 | 51.49 | 83.67 | 55.72 | 27.3 | 58.89 |
| | mean | 37.64 | 64.05 | 77.91 | 56.71 | 51.77 | 82.84 | 55.23 | 26.83 | 57.03 |
| MoLORA-2 (rank 4) - 3B | median | 39.2 | 82.14 | 80.32 | 62.5 | 50.39 | 80.58 | 57.38 | 28.47 | 60.12 |
| | mean | 38.86 | 65.71 | 80.0 | 60.0 | 50.8 | 82.17 | 56.51 | 28.03 | 57.76 |

Table 9: Comparison of MoLoRA and LoRA for the same parameter budget.

To validate that these are different approaches and outcomes, we trained a new LoRA with a rank of 40 in the T5-Large model (parameter size 770M) to compare with our MoLoRA (rank of 4, 10 experts). Additionally, for the T5-3B model parameters, we provide a comparison between MoLORA (rank of 4, 2 experts) and a single LoRA with rank 8. Results are given in Table A.5.

These results show that our MoLoRA achieves better performance compared to a single higher-rank LoRA (similarly parameter-sized) on both scales. This underscores the benefits of MoE-style multiple parallel LoRAs in handling heterogeneous fine-tuning data.

## A.6 GENERATIVE TASK PERFORMANCE

To validate our methods' efficacy in an evaluation setup that includes auto-regressive generation, we conduct an additional evaluation on a summarization dataset, namely SAMSum (Gliwa et al., 2019). Table A.6 shows the comparison between our MoV, MoLoRA, PEFT baselines ($(IA)^3$, LoRA) and full fine-tuning using the T5-3B base model. Note that this is not a held-out task, given that the training split of the dataset is included in the P3 dataset. However, we believe it may be a valuable data point for the generalization of our methods in-distribution. We calculate the rouge scores for each model.

These results confirm that MoV and MoLoRA achieve very competitive results with full fine-tuning in a more complex generative task, similar to the trend we showed in the paper's main evaluation setting.

**Comparison of Different Models**

| Base T5 3B model | Rouge1 | Rouge2 | RougeL | Avg |
|---|---|---|---|---|
| IA3 | 43.6 | 21.4 | 36.8 | 33.9 |
| MoV, 10 experts | 47.8 | 24.8 | 40.3 | 37.6 |
| MoV, 30 experts | 48.5 | 25.3 | 41.1 | 38.3 |
| LoRA, rank 4 | 46.1 | 22.4 | 38.0 | 35.5 |
| LoRA, rank 8 | 46.2 | 22.5 | 38.1 | 35.6 |
| LoRA, rank 16 | 46.3 | 23.0 | 38.5 | 35.9 |
| MoLoRA, 2 experts, rank 4 | 47.9 | 24.8 | 40.1 | 37.6 |
| Full fine-tuning (T0-3B) | 48.6 | 25.8 | 41.2 | 38.5 |

Table 10: Comparison of T5 3B model variations and their Rouge metric scores.

## A.7 CROSS-TASK TRANSFER BETWEEN TRAINING AND EVALUATION TASKS

MoE-style architectures can leverage cross-task relationships between tasks and such knowledge to achieve strong performance in unseen tasks. To validate that there are such cross-task relationships between multiple training tasks and held-out evaluation tasks, we fine-tuned task-specific $(IA)^3$ vectors for each task category in training and evaluated them for held-out tasks. As shown in Table A.7, different training tasks perform best in different held-out tasks. For example, the multiple-choice_qa task is most beneficial for super_glue_cb and super_glue_copa. Conversely, paraphrase and structure_to_text tasks work best on super_glue_rte. As validated by our experimental results, MoE-style architecture better exploits such latent relationships with expert specialization compared

to other parameter-efficient fine-tuning methods $(IA)^3$, and LoRA, resulting in higher performance with similar efficiency.

**Model Performance Metrics**

| Task | QA Closed Book | Paraphrase | QA Extractive | QA Multiple Choice | Sentiment | Structure to Text | Summarization | Topic Classification |
|---|---|---|---|---|---|---|---|---|
| CB (NLI) | 43.80 | 51.66 | 42.85 | 53.33 | 40.11 | 35.59 | 33.57 | 40.11 |
| RTE (NLI) | 57.07 | 59.81 | 52.92 | 55.30 | 53.64 | 47.25 | 47.90 | 47.97 |
| WSC (Coreference) | 45.00 | 62.40 | 40.28 | 43.84 | 43.46 | 62.30 | 61.44 | 55.96 |
| WNG (Coreference) | 49.59 | 50.19 | 51.53 | 50.93 | 49.80 | 50.19 | 49.86 | 50.22 |
| WIC (Word sense dis.) | 51.58 | 50.70 | 50.78 | 50.51 | 51.37 | 51.30 | 50.43 | 49.26 |
| HS (Sentence comp.) | 24.14 | 25.68 | 26.48 | 27.39 | 25.32 | 25.44 | 24.76 | 27.22 |
| COPA (Sentence comp.) | 49.60 | 54.59 | 51.87 | 66.95 | 57.51 | 55.15 | 53.86 | 51.98 |

Table 11: Performance metrics for different tasks on the large 770M model.

## A.8 LIMITATIONS

A primary constraint of our experimental framework is its pretrained model architecture of encoder-decoder. Based on the universality of our approach, we believe that our findings extend to decoder-only models. However, we leave this as the subject of future work. Additionally, our assessment is exclusively within the context of instruction fine-tuning. Exploration of its efficacy during the pre-training phase remains an avenue for future research.

## A.9 MoLORA IMPLEMENTATION

```
class MoLORA_Layer(nn.module):
  n_experts: int # number of experts
  rank: int # low-rank dimension
  h_out: int # output dimension

  def call(self, inputs):
    # inputs shape: [batch, seq, h_dim]
    batch, seq, h_dim = inputs.shape

    # MoLORA A: [n_experts, h_dim, rank]
    molora_A = self.param('molora_A',
      nn.init.normal(), (self.n_experts, h_dim, self.rank))

    # MoLORA B: [n_experts, rank, h_out]
    molora_B = self.param('molora_B',
      nn.init.zeros(), (self.n_experts, self.rank, self.h_out))

    # Ax: [batch, seq, n_experts, rank]
    molora_Ax = jnp.einsum('...d,edr->...er',
                           inputs,
                           molora_A)

    # BAx: [batch, seq, n_experts, h_out]
    molora_BAx = jnp.einsum('...er,ero->...eo',
                            molora_Ax,
                            molora_B)

    # router probs: [batch, seq, n_experts]
    router_probs = self.router(inputs,
      self.n_experts, dtype='float32')

    # combined LoRAs' outputs: [batch, seq, h_out]
    molora_combine = jnp.einsum('...e,...eo->...o',
                                router_probs,
                                molora_BAx)

    return molora_combine
```

