# OpenReview forum: "Pushing Mixture of Experts to the Limit: Extremely Parameter Efficient MoE for Instruction Tuning"
_ICLR.cc/2024/Conference — ICLR 2024 poster_

### Official Review · Reviewer_Bdev · 2023-10-31

**Soundness:** 2 fair
**Presentation:** 3 good
**Contribution:** 2 fair
**Rating:** 6
**Confidence:** 4

**Summary:**

The authors propose a new Parameter Efficient Fine-Tuning (PEFT) method which uses Mixture-of-Experts (MoE) architecture. In particular, the authors target two pre-existing PEFT methods which are (IA)3 and LoRA, and replace the vectors in (IA)3 and LoRA components in LoRA with MoE modules - MoV (Mixture of Vectors) and MoLoRA (Mixture of LoRA) respectively. The authors show the effectiveness of the proposed methods by comparing the downstream task fine-tuning performance. MoV and MoLoRA outperforms (IA)3 and LoRA performance and show similar performance as full weight fine-tuning. The authors also add various ablation studies including different routing strategies, varying number of experts, and expert usage on different tasks.

**Strengths:**

- The authors show that a simple combination of two well-known methods (PEFT and MoE) can achieve better quality.
- Overall, the proposed method is well described and the paper is easy to follow

**Weaknesses:**

- Training and inference efficiency analysis is missing. Table 6 briefly touched on the training of MoV case, but it would be good to have more comprehensive analysis such as memory consumption and training time. Also, LoRA has an advantage that the LoRA components can be added to the original weight and no additional inference cost is incurred. On the other hand, MoE might introduce some overheads. As this paper is about an efficient method, this kind of analysis will make the paper more comprehensive.
- Even though MoV and MoLoRA use small fraction of the original model parameters, they are using quite more parameters than (IA)3 and LoRA. This is important because we are handling quite larger models these days and those information would be useful to consider. Therefore, the information about how much memory consumption and the number of parameters will increase (not full fine-tuning experiments) would be useful to have.

**Questions:**

- 'efficiency side-effects': side-effect usually means negative, but here constant inference cost is a positive thing.
- page 2, Contributions (3): 'higly' -> 'highly'
- Do you have any insights why more number of experts doesn't always give better results?

---

> ### Author Response · Authors · 2023-11-17
>
> We appreciate **R Bdev** for their valuable feedback and their observations, noting that the “proposed method is well described and the paper is easy to follow” and that our proposed method “can achieve better quality.” We acknowledge the concern raised for the preliminary analysis of the efficiency metric being reported for our proposed method, and we used the rebuttal period to run additional, more comprehensive benchmarking
>
>
> We have now curated a comprehensive table detailing the efficiency metrics, which include parameter number, training and inference time, memory consumption, and overall performance for various methods applied to the T5-3B. By using these metrics, we aim to provide a more nuanced understanding of the performance-to-cost ratio, a crucial aspect of our study's focus on efficiency. This addition not only addresses the specific concerns raised but also enhances the overall comprehensiveness of our paper. We are confident that this addition substantially enhances the quality of our work. Furthermore, we will apply the recommended revisions addressing semantic and spelling errors in the final version of the manuscript. We respectfully request that the reviewer consider revising their evaluation in light of these improvements.
>
>
> | Method (Base model T5-XL 3B parameters) | # Params | %Decrease in Training Time w.r.t. Full FT | Increase in Inference w.r.t. Full FT (Summarization) | Peak Memory in GB (TPU v4-32) | Performance |
> | --------------------------------------- | -------- | ------------------------------------------------------------  | ------------------------------------------------------ | ----------------------------- | ----------- |
> | IA3     | 0.018%   | 38%  | 1x      | 7.88     | 52.90       |
> | MoV, 10 experts                         | 0.32%    | 31%      | 1.1x       | 10.44           | 59.93       |
> | MoV, 30 experts                         | 0.68%    | 27%     | 1.15x        | 10.62      | 60.61       |
> | LoRA, rank 4      | 0.30%    | 7%    | 1x      | 8.96        | 57.51       |
> | LoRA, rank 8    | 0.60%    | 4%      | 1x         | 9.01       | 58.89       |
> | LoRA, rank 16  | 1.20%    | 3% | 1x | 9.05  | 59.54       |
>
>
> **Training time and inference cost**: We evaluated $(IA)^3$, MoV, and LoRA for training and inference times. $(IA)^3$ trains the fastest but with the worst-performing method. MoV (10 and 30 experts) offers a balance of significantly higher training speed relative to full fine-tuning and higher performance, outperforming LoRA. In inference, $(IA)^3$ and LoRA add no extra time when their weights are integrated into the base model, while MoV increases it by only 1.1x to 1.15x depending on the number of experts.
>
> **The number of fine-tuned parameters and memory consumption**: In the table, we provide the number of fine-tuned parameters (*# Params*) for each method.  In terms of updated parameters vs. performance trade-off, our MoV method outperforms others while balancing parameter count and performance effectively. Specifically, MoV with 30 experts surpasses LoRA rank 16 in performance while updating a smaller number of parameters.
>
> We also include the maximum memory usage during training for each method.  Note that this memory usage is influenced by the total parameters, fine-tuned number of parameters, optimizer, batch size, and hardware. Notably, our MoV method at the 3B scale offers optimal performance without significant efficiency costs during training and inference. It also scales effectively to 11B, maintaining competitive fine-tuning performance, as shown in Figure 1 and Table 4. We plan to include this table and a detailed cost-benefit analysis in the final paper in response to reviewer feedback.
>
> > Do you have any insights why more number of experts doesn't always give better results?
>
> We hypothesize that as the number of experts increases, each additional expert contributes less to overall performance improvement because the most significant gains are often achieved early on when key expertise areas are covered. Additional experts might overlap in their specialization, leading to diminishing returns; this notion is reflected in the routing probabilities of experts in Figure 5. The same empirical observations were illustrated in the following [1, 2].
>
> [1] Fedus, William, et al. “Switch Transformers: Scaling to Trillion Parameter Models with Simple and Efficient Sparsity.” arXiv preprint arXiv:2101.03961 (2022).
>
> [2] Shen, Sheng, et al. "Mixture-of-experts meets instruction tuning: A winning combination for large language models." arXiv preprint arXiv:2305.14705 (2023).

---

> > ### Author Response · Authors · 2023-11-19
> >
> > Now that the discussion is underway, we wanted to ask **R Bdev** if there are any follow-up points we can clarify.
> >
> > We would like to add a further analysis that we conducted regarding the number of experts at inference: During the inference time, we used only the top-K experts that are determined by the router probabilities to decrease the inference cost. We found that when a large number of experts are trained (e.g., 30), using only a top-K (e.g., 15) subset of these experts during inference maintains high performance. Concretely, our MoV-30 with top-15 experts achieves an average performance of 60.25. This is very competitive with using all experts (60.61); however, it decreases the inference latency by 1%.
> >
> > Note that MoV-30 trains 1.3x faster than similar parameter-size LoRA (rank 8) and outperforms LoRA by 1.4 (top-15) - 1.7 (all 30 experts) average performance.
> >
> > If there are no further points of clarification regarding the manuscript and extended performance-efficiency trade-off analysis, we would ask that reviewer **R Bdev** consider increasing their score to reflect.

---

> > > ### Comment · Reviewer_Bdev · 2023-11-21
> > >
> > > Thanks for providing additional studies in the rebuttal.
> > > I think this improves the quality of the paper.
> > > I will change my rating to 6 based on that changes the authors are making.

---

> > > > ### Author Response · Authors · 2023-11-21
> > > >
> > > > Dear reviewer, **Bdev**,
> > > >
> > > > Thank you for raising your score in light of the clarifications to the manuscript and experiments on the efficiency metrics of our method. We are glad that we were able to address your concerns and are grateful for your valuable suggestions, which provided us the opportunity to clarify the significance of our results and run additional experiments confirming our insights on the efficiency of our approach concerning memory usage, inference, and training reduction. We would like to thank the reviewer again for their positive update and detailed feedback on the presentation of the current manuscript. Please let us know if there are any other concerns or questions we can address during the rebuttal period, and we would be happy to clarify.

---

### Official Review · Reviewer_hesi · 2023-10-31

**Soundness:** 3 good
**Presentation:** 3 good
**Contribution:** 2 fair
**Rating:** 5
**Confidence:** 5

**Summary:**

This paper aims at conducting efficient instruction tuning with language models equipped with mixture of expert layers, while to manage the large computational cost, the authors propose to consider each single copy of PEFT parameters as an expert instead of a copy of MLP, the common definition of experts, for better usage. Extensive experiments demonstrate the superiority of the proposed MOV and MOLORA architectures.

**Strengths:**

- The authors utilize extensive accuracy vs. compute figures to demonstrate the superior efficiency.
- The authors emphasize the scalability of the proposed methods, which is important for large models.
- The paper writing and architecture is clear with easy to read.

**Weaknesses:**

- About expert specialization and load balancing:
  - One of the fundamental reasons to utilize MoE-style architecture is expert specialization.
  - In Sec. 3, the authors mention to apply load balance loss specifically for discrete merging. Is load balancing applied also for soft merging? And does soft merging rely on load balancing loss to prevent expert collapse?
  - If not, how to specilize the abilities of each expert to obtain the results in Fig. 5 with simple token-level routing without any explicit control?
  - If indeed we can get specialized experts, does the similar distribution shown in Fig. 5 suggest that the evaluated tasks still have connection with the several training tasks, otherwise it is hard to understand why a linear router can generate such a low entropy distribution on the evaluated tasks with a zero-shot setting?
  - In other words, if the evaluated tasks differ from the training tasks a lot, the reasonable routing distribution should be close to uniform, especially for a linear router.
- About parameter efficient fine tuning (PEFT):
  - Another way to view this work is another way to increase amounts of PEFT parameters.
  - But one perspective of current development of PEFT is that the amount of parameters is the most important, while how you deploy the parameters does not make much differ.
  - In Tab. 1, comparing LoRA (r=4) & MoV-10, LoRA (r=8) & MoV-30 and LoRA (r=16) & MoV-60 (all pairs with similar learnable parameters), the performance gaps are 2.42%, 1.72% and 0.29%, greatly shrinking with respect to the number of parameters, doubting about the scalability of the proposed method.
- Overall,
  - From the perspective of MoE, this paper shares similar architecture with [1], and unfortunately still cannot answer the basic question including why the specialized architecture can benefit instruction tuning on novel tasks.
  - From the perspective of PEFT, the methods can still not convince me that it is non-trivial to other PEFT methods by simply adding more parameters for better results.

[1] Shen, Sheng, et al. "Mixture-of-experts meets instruction tuning: A winning combination for large language models." *arXiv preprint arXiv:2305.14705* (2023).

**Questions:**

- Implementation details:
  - Soft vs. discrete merging: There are two things to decide, including how many experts to choose and how to combine these experts. According to the equation in Sec. 2.2, does soft merging suggest that you first ensemble different expert weights according to the router output and then conduct forward propagation for only once? If so, does this manner only apply for (IA)3 or for LoRA with discrete merging also, since the previous manner is different from our common modeling of MoE architecture? Would you mind further explain how you conduct this ablation?
  - What is the "average median score", the main evaluation metric used by the authors, first mentioned in the first line of Page 6?

[1] Shen, Sheng, et al. "Mixture-of-experts meets instruction tuning: A winning combination for large language models." *arXiv preprint arXiv:2305.14705* (2023).

---

> ### Author Response · Authors · 2023-11-17
>
> We would like to thank **R hesi** for their detailed review. We are particularly flattered that they mentioned we “utilize extensive accuracy vs. compute figures to demonstrate the superior efficiency.” Also, we appreciate them noting the “scalability of the proposed methods, which is important for large models.” We address their remaining concerns below:
>
> > In Sec. 3, the authors mention to apply load balance loss specifically for discrete merging. Is load balancing applied also for soft merging? And does soft merging rely on load balancing loss to prevent expert collapse? If not, how to specilize the abilities of each expert to obtain the results in Fig. 5 with simple token-level routing without any explicit control?
>
> Thanks for pointing out an unclear description. Our soft merging does not rely on any external load-balancing loss. The main reason that simple soft merging achieves expert specialization is the distinct task and dataset composition in the training, which is characterized by instruction tuning. Since we start with a pre-trained model, token representations are already diversified from each other based on different tasks and datasets, enabling routing to be specialized.
>
> In our experiments, we found that load balancing loss decreases performance in both soft and discrete routing cases. We attribute this to two potential factors: (1) highly different data sizes for each dataset in our instruction tuning mixture, and (2) the misalignment between pretraining loss and fine-tuning loss when we add an auxiliary load balancing loss. This is similar to the experimental result shown in [1; Section 4.1, FLAN-ST_base].
>
>
> > If indeed we can get specialized experts, does the similar distribution shown in Fig. 5 suggest that the evaluated tasks still have connection with the several training tasks, otherwise it is hard to understand why a linear router can generate such a low entropy distribution on the evaluated tasks with a zero-shot setting?
>
> Thanks for giving us the opportunity to highlight this important point. Indeed, there are cross-task relationships between multiple training tasks and held-out evaluation tasks. For example, [2] has shown that question-answering tasks such as “cosmos_qa” and “quail” have strong cross-task transfer to NLI tasks (e.g., super glue cb, super glue rte).
>
> To validate this, during the rebuttal period, we fine-tuned task-specific $(IA)^3$ vectors for each task category in training and evaluated them for held-out tasks. As shown in the table below, different training tasks perform best on different held-out tasks. For example, the multiple-choice_qa task is most beneficial for super_glue_cb and super_glue_copa. Conversely, paraphrase and structure_to_text super_glue_rte tasks work best on super_glue_wsc.
>
> | Task     | (T-few) QA Closed Book | (T-few)  Paraphrase Task | (T-few)  QA Extractive | (T-few) QA Multiple Choice | (T-few) Sentiment | (T-few) Structure to Text | (T-few)  Summarization | (T-few)  Topic Classification |
> | ------------------------------------- | -------------------------------------- | --------------------------------------- | ------------------------------------- | ------------------------------------------ | --------------------------------- | -------------------------------------------- | ------------------------------------ | ------------------------------------------------ |
> | NLI: super_glue - cb                  | 43.8    | 51.6                                  | 42.8    | 53.3  | 40.1       | 35.5     | 33.5   | 40.1   |
> | NLI: super_glue - rte                 | 57.0    | 59.8        | 52.9        | 55.3         | 53.6                            | 47.2     | 47.9   | 47.9                    |
> | Coreference: super_glue - wsc.fixed   | 45.0   | 62.4    | 40.2    | 43.8     | 43.4   | 62.3      | 61.4       | 55.9           |
> | Coreference: winogrande - winogrande.xl | 49.5 | 50.1     | 51.5      | 50.9   | 49.8    | 50.1     | 49.8     | 50.2 |
> | Word sense disambiguation: super_glue - wic | 51.5  | 50.7  | 50.7   | 50.5     | 51.3                            | 51.3   | 50.4    | 49.2                     |
> | Sentence completion: hellaswag        | 24.1 | 25.6      | 26.4   | 27.3       | 25.3 | 25.4     | 24.7      | 27.2   |
> | Sentence completion: super_glue - copa | 49.6  | 54.5    | 51.8   | 66.9   | 57.5      | 55.1   | 53.8         | 51.9                       |
>
> As validated by our experimental results, MoE-style architecture better exploits such latent relationships with expert specialization compared to other parameter-efficient fine-tuning methods $(IA)^3$, and LoRA, resulting in higher performance with similar efficiency.

---

> > ### Author Response · Authors · 2023-11-17
> >
> > > In Tab. 1, comparing LoRA (r=4) & MoV-10, LoRA (r=8) & MoV-30 and LoRA (r=16) & MoV-60 (all pairs with similar learnable parameters), the performance gaps are 2.42%, 1.72% and 0.29%, greatly shrinking with respect to the number of parameters, doubting about the scalability of the proposed method.
> >
> >
> > We appreciate you asking this question. An important clarification is that there is a natural ceiling to performance—even the full fine-tuning, which should be considered as the ceiling of the baseline comparisons, improves by 4.43% over LoRA (r = 4), 1.98% over LoRA (r = 8), and 0.87% over LoRA (r = 16). This also shrinks relative to LoRA variants, even though it involves all parameters. Hence, while the gap shrinks for our proposed methods, it is still a huge improvement in efficiency over full-finetuning.
> >
> > Furthermore, the MoV-10, which updates around the same number of parameters as LoRA (r = 4), is 1.3x (~22%) faster than LoRA rank 4 in addition to a 2.42 performance increase. This stark contrast between our proposed MoV and LoRA (the most widely used PEFT techniques) shows how MoV excels at effectively balancing performance improvement with training efficiency.
> >
> > Also worth noting that at the 3B scale, our MoV with 30 experts achieves the best result among our variants, achieving a 1.07 performance increase compared to LoRA rank 16 while fine-tuning a smaller number of parameters (0.68% vs. 1.2%). On a side note, in our response to reviewer #4 (Bdev), we explain potential diminishing returns as we increased the number of experts as an empirical observation we found similar to previous work [1, 3].
> >
> > > From the perspective of MoE, this paper shares similar architecture with [1], and unfortunately still cannot answer the basic question including why the specialized architecture can benefit instruction tuning on novel tasks.
> >
> > Our approach significantly differs from [1] as we offer a novel framework, unifiying parameter-efficient fine-tuning (PEFT) with MoE-style routing by the use of lightweight adapters like $(IA)^3$ vectors instead of the feed-forward component of every other Transformer layer. Our methods maintain high performance without increasing the model size and without updating a large number of parameters. This is particularly important when comparing our method with a standard MoE architecture, as in [1], because standard MoE increases the number of parameters, hence the model size, to a very large scale (e.g., FLAN-XXL 395B parameters), where fine-tuning these models is not possible with limited compute memory resources. Our approach, however, emphasizes efficiency and flexibility in adapting existing model weights (e.g., T5-XXL 11B).
> >
> > Specialized MoE-style architecture is beneficial in the context of instruction fine-tuning. In the table shown above, by fine-tuning each single vector for specific tasks, we notice PEFT vectors fine-tuned only for specific tasks show a stronger zero-shot evaluation score relative to specific held-out tasks. Unlike instruction tuning with a single PEFT adapter, our MoE-style allows for ongoing learning of separate training tasks without the risk of catastrophic interference and exhibits compositional abilities through the soft merging of various experts via a router.
> >
> > > From the perspective of PEFT, the methods can still not convince me that it is non-trivial to other PEFT methods by simply adding more parameters for better results.
> >
> > Our empirical analysis of the T5-3B model reveals that our mixture style of vectors, such as MoV-30, utilizing only 0.68% of the parameters, achieves a performance score of 60.61. In contrast, LoRA (with a rank of 16) updates nearly double the parameters (1.2%) but attains a lower performance of 59.54. Notably, MoV-30 is also 1.3x faster than LoRA (rank 16).
> >
> > Also, as suggested by reviewer #1 (s33v), we conducted additional experiments during the rebuttal period comparing MoLoRA and LoRA (with a higher rank). Results show that MoLoRA (rank 4, 10 experts) outperforms similar parameter-sized LoRA (rank 40) for the T5-770M model. Similarly, MoLoRA (rank 4, 2 experts) performs better than both rank 8 and rank 16 LoRA for T5-3B, validating the effectiveness of our parameter-efficient MoE-style method.

---

> > > ### Author Response · Authors · 2023-11-17
> > >
> > > > Soft vs. discrete merging: There are two things to decide, including how many experts to choose and how to combine these experts. According to the equation in Sec. 2.2, does soft merging suggest that you first ensemble different expert weights according to the router output and then conduct forward propagation for only once? If so, does this manner only apply for (IA)3 or for LoRA with discrete merging also, since the previous manner is different from our common modeling of MoE architecture? Would you mind further explain how you conduct this ablation?
> > >
> > > In the case of MoV, soft merging corresponds to an ensemble of $(IA)^3$ vectors with router weights before the forward propagation [REF to eq. and implementation]. However, in the MoLoRA, in order to benefit from the memory advantages of LoRA, we merge the LoRA’s outputs after a forward propagation of each LoRA [Section 2.2, page 4].
> > >
> > > For the discrete routing ablations, we keep the same computation as explained above since, in the case of MoV, the computation of soft merging $(IA)^3$ weights is exactly the same as merging $(IA)^3$ outputs.
> > >
> > > > What is the "average median score", the main evaluation metric used by the authors, first mentioned in the first line of Page 6?
> > >
> > > Each evaluation dataset, along with the training datasets, comprises multiple prompt templates. For example, the Super Glue Copa dataset used for evaluation contains 12 prompt templates per data point. We median performance for each evaluation dataset across the prompt templates. Here, the term "average median score" denotes the average of these per-dataset results. We report the median score to be consistent with our baseline, T0 [4], but we also provide mean accuracy scores per dataset in the Appendix. We will clarify this in the manuscript.
> > >
> > >
> > > [1] Shen, Sheng, et al. "Mixture-of-experts meets instruction tuning: A winning combination for large language models." arXiv preprint arXiv:2305.14705 (2023).
> > >
> > > [2] Zhou, Jing, et al. "Not All Tasks Are Born Equal: Understanding Zero-Shot Generalization." The Eleventh International Conference on Learning Representations. 2022.
> > >
> > > [3] Fedus, William, et al. “Switch Transformers: Scaling to Trillion Parameter Models with Simple and Efficient Sparsity.” arXiv preprint arXiv:2101.03961 (2022).
> > >
> > > [4] Sanh, Victor, et al. “Multitask Prompted Training Enables Zero-Shot Task Generalization” arXiv preprint arXiv:2110.08207 (2022).

---

> > > > ### Author Response · Authors · 2023-11-22
> > > >
> > > > As the discussion period is nearing its end, we wanted to ask **R hesi** if there are any follow-up points we can clarify. We have responded to all concerns raised, in addition to taking a few days to run the extensive experiments necessary to demonstrate our points, all of which are incorporated in the appendix of the manuscript. If there are no further points of clarification regarding the manuscript, expert specialization, cross-task relationships, and the difference between our method and the standard MoE, we would kindly ask that reviewer hesi to consider increasing their score to reflect

---

> ### Comment · Reviewer_hesi · 2023-11-23
> **Response to Author Rebuttal**
>
> Thanks for the detailed response. However, there are several questions raised in my initial review that the authors do not response directly.
>
> **1. About the usage of loading balancing loss.**
>
> The authors specifically claim that they do not adopt loading balancing loss, suggesting that the learnt experts can not be **sparsely** activated during training, otherwise according to [1], even initialization with a pre-trained model would still meet the risk of the expert degradation problem.
>
> **2. About the performance ceiling for fully fine-tuning.**
>
> If this so called **performance ceiling** does exist and 1) it is close with the performance of full fine-tuning, how to explain why in Table 1 MoLORA-15 (rank 4) exceeds the T0-3B baseline? 2) If this ceiling is way higher than the performance of full fine-tuning (and also MoLORA), then the authors' response do not make sense any more.
>
> **3. About the novelty of the architecture.**
>
> Instead of a novel architecture, the proposed method is just an another implementation of the tranditional MoE framework. The tranditional MoE is defined in the Equation (MoE) on top of Page 4 in the main paper, and there are lots of different implementation for the $E_i(\cdot)$ function, including 1) MLPs, 2) IAs and 3) LoRAs.
>
>
> Therefore, I do agree with Reviewer EZ7y that MoE is utilized in this paper only because the authors insist on using it without a solid research motivation, and as shown in the experiment tables in the authors' reply to Reviewer EZ7y, the less than **2.0%** improvement is also far
> away from **"Pushing Mixture of Experts to the Limit"** in the paper title. Thus, my score stands.
>
> [1] Wu, Lemeng, et al. "Residual mixture of experts." arXiv preprint arXiv:2204.09636 (2022).

---

### Official Review · Reviewer_EZ7y · 2023-10-31

**Soundness:** 3 good
**Presentation:** 3 good
**Contribution:** 3 good
**Rating:** 8
**Confidence:** 4

**Summary:**

The authors propose a parameter-efficient Mixture of Experts (MoE) training approach. By combining MoE architecture with lightweight experts, the paper introduces Mixture of Vectors (MoV) and Mixture of LORA (MoLORA), optimized for parameter efficiency training.  In particular, each expert is replaced with a lightweight PEFT adapter such as $(IA)^3$ vectors or LORA adapters. The proposed MoV and MoLORA, are highly efficient in terms of parameters. By updating less than 1% of the model’s parameters, MoV and MoLORA consistently maintain higher performance compared to standard PEFTs and achieve comparable results with full fine-tuning.

**Strengths:**

The paper is well-written, presenting a clear and sound approach. The subject matter is highly relevant to the ML community, addressing a critical challenge: while MoEs holds great potential, its practical application has been notably hindered by prohibitive computational expenses and training instabilities, rendering it inaccessible to many researchers. Showing that parameter-efficient methods such as $(IA)^3$ or LORA can substantially improve the feasibility and effectiveness of training MoEs is a significant and valuable contribution to the field.

The authors have provided extensive ablation studies, systematically evaluating the effectiveness of their proposed MoV and MoLORA approaches against parameter-efficient fine-tuning (PEFT) strategies, and full fine-tuning. The evaluations cover multiple model sizes from the T5-family, adapter types, the number of experts, and routing mechanisms.

**Weaknesses:**

- All experiments in this paper have been conducted exclusively on T5 model family (encode-decoder architecture). Showing that the proposed light-weight MoEs additionally works for decoder-only architectures would significantly strengthen the paper’s contributions and findings.

- The zero-shot evaluation tasks considered in this paper are mainly classification/multiple choice selection tasks and require generating a single token. The paper does not clearly articulate the adaptability of the proposed method to more complex tasks, such as summarization, translation, or coding, which necessitate the auto-regressive generation of longer sequences. It would be beneficial to explore and elucidate how the experts and routers operate in such long-seq generation scenarios.

**Questions:**

The authors show that token routing performs better than sentence routing across various model sizes. An alternative input to the router could be the representation derived from the last encoder layer of T5. How does this encoded representation perform in comparison? This could also offer considerable computational advantages, particularly in situations involving a large number of experts, as it would allow for the pre-computation of the weighted average of experts immediately following the generation of the encoder representation. On the other hand, it would be interesting to see if adding or concatenating the encoder's representation to the representation of tokens further improve performance.

---

> ### Author Response · Authors · 2023-11-16
>
> We thank **R EZ7y** for their very positive review of our work, noting that our paper is “well-written, presenting a clear and sound approach” and emphasizing strongly the importance of the research problem. We greatly appreciate their claim that our proposed work is a “significant and valuable contribution to the field.” We conducted an additional experiment during the rebuttal period to evaluate our method on additional summarization tasks.
>
> >Regarding the use of the T5 model family in our experiments and extending to decoder-only architectures:
>
> We thank **R EZ7y** for suggesting an additional experimental axis to further evaluate our proposed method. We note that our choice to focus on the T5 model family was partly dictated by how extensive and expensive our experiments were. We trained 100+ models, the majority of them on 3B and 11B scales, including preliminary experiments, final model runs, ablations, and analysis. T5 is also heavily used in prior work [1, 2], which allows for a clean comparison.
>
> Our primary focus in the given work is to show the effectiveness of our MoE-style parameter-efficient architecture and propose an efficient alternative to the full fine-tuning baseline for instruction tuning. However, we agree that an important direction of future work is extending these results to other architectures.
>
> > Regarding our evaluation setup and evaluating our proposed methods on more complex auto-regressive generation tasks:
>
> We clarify that we chose these tasks to fairly compare with T0 by mirroring its exact fine-tuning and evaluation setup, thereby showcasing our method's relative efficacy against a key benchmark in instruction-based tuning. These tasks are the comprehensive set of prompt instructions from the **(P3), so we also avoid any bias introduced by selecting a subset that favors our method**. We follow the same procedure as [4] where each task is converted into the format provided by the templates [1]
>
> However, to validate our methods’ efficacy in an evaluation setup that includes auto-regressive generation, we conduct an additional evaluation during the rebuttal period on a summarization dataset, namely SAMSum [3]. Below, we compare our MoV and MoLoRA with PEFT baselines ($(IA)^3$, LoRA) and full fine-tuning using the T5-3B base model. Note that this is not a held-out task, given that its training split is included in the P3 dataset. However, we believe it may be a valuable data point for the generalization of our methods in-distribution. We calculate the rouge scores for each model.
>
> The below results confirm that MoV and MoLoRA achieve very competitive results with full fine-tuning in a more complex generative task, similar to the trend we showed in the paper’s main evaluation setting.
>
> | Base T5 3B model     | Rouge1 | Rouge2 | RougeL | Avg   |
> | -------------------- | ------ | ------ | ------ | ----- |
> | $(IA)^3$   | 43.6   | 21.4   | 36.8   | 33.9  |
> | MoV, 10 experts      | 47.8   | 24.8   | 40.3   | 37.6  |
> | MoV, 30 experts      | 48.5   | 25.3   | 41.1   | 38.3  |
> | LoRA, rank 4         | 46.1   | 22.4   | 38.0   | 35.5  |
> | LoRA, rank 8         | 46.2   | 22.5   | 38.1   | 35.6  |
> | LoRA, rank 16        | 46.3   | 23.0   | 38.5   | 35.9  |
> | MoLoRA, 2 experts, rank 4 | 47.9 | 24.8 | 40.1 | 37.6  |
> | Full fine-tuning (T0-3B) | 48.6 | 25.8 | 41.2 | 38.5  |
>
> >Regarding further exploration of routing input, particularly with the use of representation derived from the last encoder layer of T5:
>
> In this work, we limit our exploration in terms of routers’ input by only comparing pre-computed sentence embeddings and token representation. We are heartened that **R EZ7y** acknowledges we have already conducted extensive experiments (doubling the number of variants) to compare token routing to sentence routing. Our results show token routing achieves better performance since routers can exploit similarities between tokens for expert decisions, offering a better generalization. This is at least a preliminary signal that a higher degree of inductive bias for task datasets (the representation derived from the last encoder layer would take this even further) is not necessarily beneficial, as one can acquire a diverse set of task knowledge directly from the hidden representations of tokens. We leave further exploration to future work.
>
> [1] Sanh, Victor, et al. “Multitask Prompted Training Enables Zero-Shot Task Generalization” arXiv preprint arXiv:2110.08207 (2022).
>
> [2] Liu, Haokun, et al. "Few-shot parameter-efficient fine-tuning is better and cheaper than in-context learning." Advances in Neural Information Processing Systems 35 (2022): 1950-1965.
>
> [3] Gliwa, Bogdan, et al. "SAMSum corpus: A human-annotated dialogue dataset for abstractive summarization." arXiv preprint arXiv:1911.12237 (2019).
>
> [4] Colin, Raffel, et al. “Exploring the Limits of Transfer Learning with a Unified Text-to-Text Transformer.” arXiv preprint arXiv:1910.10683 (2020).

---

> > ### Comment · Reviewer_EZ7y · 2023-11-22
> >
> > I would like to thank the authors for their responses. I am happy with the contributions of the paper and will keep the rating of 8.

---

> > > ### Author Response · Authors · 2023-11-22
> > >
> > > Dear Reviewer **EZ7y**,
> > >
> > > We sincerely appreciate your decision to preserve your score following our recent clarifications and additional experiments to demonstrate our proposed method's ability to extend our zero-shot evaluation. Your constructive feedback has been invaluable in enhancing the clarity and impact of our manuscript. We are particularly grateful for the opportunity you provided to further demonstrate the significance of our findings. The additional experiments have reinforced our insights regarding the method's approach to solving complex auto-regressive generation tasks. Thank you once again for your positive reassessment and the detailed feedback you have provided on the presentation of our work. If there are any further questions or concerns you may have during the rebuttal period, please do not hesitate to let us know. We are more than willing to provide any necessary clarifications.

---

### Official Review · Reviewer_s33v · 2023-11-01

**Soundness:** 3 good
**Presentation:** 3 good
**Contribution:** 2 fair
**Rating:** 8
**Confidence:** 4

**Summary:**

The paper addresses how to leverage MoEs for instruction fine-tuning and presents MoV and MoLORA. The use of MoE in fine-tuning leverages the fact that conditional computation is efficient while avoiding the disadvantage that MoE requires relatively large storage by combining MoE with (IA)^3 and LoRa. In-depth experimental studies demonstrate that the proposed method achieves competitive results with full fine-tuning, only 1% of the parameters involved, and no limitation to the model scale, and can also adapt the model to unseen tasks.

**Strengths:**

1.	Leverage the advantages and avoid the disadvantages of the MoE structure when combined with PEFT methods.
2.	In-depth ablation study demonstrates the capabilities and scalabilities of the proposed method.

**Weaknesses:**

1.	MoE takes advantage of the inherent heterogeneity of the data, allowing different parameters to handle different distributions in the data. However, fine-tuning usually focuses on one or a few downstream tasks. In this case, the motivation for using ten or even dozens of experts for learning requires further justification.
2.	LoRa exploits the inherent low-rank properties of large models. However, multiple parallel LoRa matrices are mathematically equivalent to higher rank LoRa. The effectiveness of MoLoRa proposed in this article cannot be proven without comparison with higher-rank lora.

**Questions:**

The article mentioned that the larger the batch size, the easier it is for MoE to collapse to an expert. However, the difference between fine-tuning and pretrain is that the model has entered a relatively stable and better-performing landscape before training begins. Why under such conditions, the more common gradient direction brought by a larger batchsize will still cause a certain expert to dominate the gradient descent direction of the parameter space?

---

> ### Author Response · Authors · 2023-11-15
>
> We would like to thank **R s33v** for their positive feedback and for highlighting the “in-depth ablation study” that shows the “capabilities and scalabilities'' of our proposed method of combining PEFT with MoEs. We also express gratitude towards them for emphasizing positively that our approach, “leverage the advantages and avoid the disadvantages of the MoE structure when combined with PEFT methods.” We address some of the individual concerns below:
>
> > MoE takes advantage of the inherent heterogeneity of the data, allowing different parameters to handle different distributions in the data. However, fine-tuning usually focuses on one or a few downstream tasks. In this case, the motivation for using ten or even dozens of experts for learning requires further justification.
>
> **R s33v** is indeed correct that if there were only one or a few downstream tasks, we wouldn’t use as many experts. However, we are interested in the multi-task fine-tuning setting that has led to recent breakthroughs in NLP [1], where the fine-tuning dataset is deliberately structured to represent a **diverse and large** set of tasks, with a high degree of heterogeneity in the data. Specifically in our experimental setup, the P3 dataset we use for fine-tuning **contains 62 datasets, each containing data points with 5 to 20 unique prompt templates, where there are 8 distinct tasks** such as paraphrasing, QA closed book, QA extractive, QA multiple-choice, sentiment, summarization, topic classification, and word-sense disambiguation. Hence, multiple experts are more appropriate here to gain from the benefits of modular specialization given the diversity in the number of datasets, prompts, and tasks.
>
> > The article mentioned that the larger the batch size, the easier it is for MoE to collapse to an expert. However, the difference between fine-tuning and pretrain is that the model has entered a relatively stable and better-performing landscape before training begins. Why under such conditions, the more common gradient direction brought by a larger batch size will still cause a certain expert to dominate the gradient descent direction of the parameter space?
>
> We clarify to **R s33v** that our setting differs from “traditional” finetuning where all weights have already been calibrated during pretraining. Even though the model is indeed pre-trained, both the router layers and PEFT experts are **randomly initialized before finetuning** which increases instability during finetuning relative to updating all pre-trained weights.
>
> Our finding that larger batch sizes lead to the learning collapse in MoE is consistent with other findings that MoE models benefit from smaller batch sizes due to the instability of training and fine-tuning MoE-style architectures using large batch sizes [2, 3]. We highlight this finding to contribute to the understanding of limitations for MoE approaches.
>
> [1] Sanh, Victor, et al. “Multitask Prompted Training Enables Zero-Shot Task Generalization” arXiv preprint arXiv:2110.08207 (2022).
> [2] Zoph, Barret, et al. “ST-MoE: Designing Stable and Transferable Sparse Expert Models.” arXiv preprint arXiv:2202.08906 (2022).
> [3] Shen, Sheng, et al. "Mixture-of-experts meets instruction tuning: A winning combination for large language models." arXiv preprint arXiv:2305.14705 (2023).

---

> > ### Author Response · Authors · 2023-11-15
> >
> > In response to **R s33v**, we have also run additional experiments during the rebuttal and will update the manuscript accordingly. We are happy to continue to engage further if the answers below are insufficient and ask that **R s33v** consider updating their score if satisfactory. Please let us know if something else needs further clarification or if we missed something.
> >
> > > LoRa exploits the inherent low-rank properties of large models. However, multiple  parallel LoRa matrices are mathematically equivalent to higher rank LoRa. The effectiveness of MoLoRa proposed in this article cannot be proven without comparison with higher-rank lora.
> >
> > We thank **R s33v** for this opportunity to strengthen our work. We take this opportunity during rebuttals to **run additional benchmarking experiments at scale (770M) and also reference the 3B model experiment from Table 3** to compare MoLoRA to higher-rank LoRA.
> >
> > It is important to clarify that multiple parallel LoRA in our MoE-style architecture differ **in terms of optimization** from single higher-rank LoRA **due to the use of routers**. By conditioning via the router that is sensitive to different input data, small LoRA weights in the MoLoRA can exploit different low-rank properties. Therefore, MoLoRA trained with MoE-style routing enables better “specialization” for different tasks in each smaller LoRA.
> >
> > Unlike MoLoRA, a single higher-rank LoRA that lacks such router conditioning exploits general properties already emphasized in the base model [1; Appendix H.3]. This limits specialization and can be suboptimal, especially when fine-tuning data is highly heterogeneous.
> >
> > To validate that these are different approaches and outcomes, based on **R s33v’s** suggestion, we trained a new LoRA with a rank of 40 in the T5-Large model (parameter size 770M) to compare with our MoLoRA (rank of 4, 10 experts). Additionally, for the T5-3B model parameters, we provide a comparison between MoLORA (rank of 4, 2 experts) and a single LoRA with rank 8. We will update the manuscript accordingly to reflect these experiments. Both results are given below.
> >
> > | Methods for T5-Large (770M) | Num Params | ANLI | CB  | RTE | WSC | WIC  | COPA | WNG | HS  | AVG  |
> > | --------------------------- | ---------- | ---- | --- | --- | --- | ---- | ---- | --- | --- | ---- |
> > | LoRA (rank 40)                | 5%         | 35.7 | 64.3| 58.8| 63.5| 50.3 | 78.5 | 52.9| 26.8| 53.9 |
> > | MoLoRA-10 (rank 4)   | 5.6%       | 33.2 | 67.9| 68.4| 64.9| 50.4 | 80.0 | 52.6| 27.2| 55.5 |
> >
> > | Methods for T5-XL (3B) | Num Params | ANLI | CB  | RTE | WSC | WIC  | COPA | WNG | HS  | AVG  |
> > | ---------------------- | ---------- | ---- | --- | --- | --- | ---- | ---- | --- | --- | ---- |
> > | LoRA (rank 8)            | 0.6%       | 37.5 | 75.0| 78.0| 62.5| 51.5 | 83.7 | 55.7| 27.3| 58.9 |
> > | MoLoRA-2 (rank 4)  | 0.75%    | 39.2 | 82.1| 80.3| 62.5| 50.4 | 80.6 | 57.4| 28.5| 60.1 |
> >
> > These results show that our MoLoRA achieves better performance compared to a single higher-rank LoRA (similarly parameter-sized) on both scales. This underscores the benefits of MoE-style multiple parallel LoRAs in handling heterogeneous fine-tuning data.
> >
> > [1] Hu, Edward J., et al. “LoRA: Low-Rank Adaptation of Large Language Models.” arXiv preprint arXiv:2106.09685 (2021)

---

> > > ### Author Response · Authors · 2023-11-17
> > >
> > > Now that the discussion is underway, we wanted to ask **R s33v** if there are any follow-up points we can clarify. If there are no further points of clarification regarding the manuscript and the justification for using multiple experts in MoE given its focus on data heterogeneity, the comparison of our proposed MoLoRa with higher-rank LoRa, and the exploration of how larger batch sizes influence expert dominance, we would ask that reviewer **R s33v** consider increasing their score to reflect the improvements and clarifications we have provided. We are very happy to continue to engage and answer any questions.

---

### Author Response · Authors · 2023-11-15

We greatly appreciate the thoughtful and positive feedback from the reviewers. A critical contribution of our work is that we navigate a severe constraint – our aim is to update only a small percentage of all parameters while also navigating the optimization challenges inherent to MoEs. As described by **EZ7y** “while MoEs holds great potential, its practical application has been notably hindered by prohibitive computational expenses and training instabilities, rendering it inaccessible to many researchers."

We are encouraged that reviewers found the results notable “Extensive experiments demonstrate the superiority of the proposed MOV and MOLORA architectures” **[hesi]**. “Showing that parameter-efficient methods such as $(IA)^3$ or LORA can substantially improve the feasibility and effectiveness of training MoEs is a significant and valuable contribution to the field” **[EZ7y]**. Reviewers have favorably evaluated our experimental setup, commending its clarity and comprehensiveness, while also praising the scalability of our approach for its adaptability and potential in larger models **[hesi, s33v, Bdev]**. “The authors utilize extensive accuracy vs. compute figures to demonstrate the superior efficiency”

We are heartened by the reviewers' acknowledgment of the extensive experimental setup of our work “The authors emphasize the scalability of the proposed methods, which is important for large models”  and “provided extensive ablation studies, systematically evaluating the effectiveness of their proposed MoV and MoLORA approaches against parameter-efficient fine-tuning (PEFT) strategies, and full fine-tuning” **[s33v, EZ7y, Bdev, hesi]**.

We also thank the reviewers for their individual constructive feedback. We have been using this rebuttal time to run additional experiments, and because of the scale, it has taken a few days to gather the additional results. However, we expect to be able to respond to individual reviewers very shortly, and we will also be updating the manuscript as we respond. We look forward to engaging in meaningful discussion and welcome any additional questions at any point in time.

---

### Author Response · Authors · 2023-11-21

After carefully reading comments and suggestions from all reviewers, we have made the following new experimental contributions and changes over the course of the rebuttal period, in addition to meticulously responding to all the reviewer's concerns:

* We've upgraded **Table 6** (the updated version is in **Appendix A.4**) with additional training time metrics for LoRA methods. Moreover, we included new tables (**Tables 7 & 8**) for peak memory usage and various inference metrics:
    * For fine-tuning, our MoV trains faster than LoRA. For inference, it introduces only a small increase in latency for large sequence lengths.
    * These enhancements stem from gathering more comprehensive data, offering a deeper insight into the benefits of various methods.
* In our commitment to rigor and depth, we conducted extensive experiments, yielding a detailed and insightful analysis of the differences between LoRA and MoLoRA for a better understanding of the nuances that set these methods apart. These updates are reflected in **Appendix A.6**.

* Moreover, we undertook further experiments to calculate the rouge scores for both our proposed method and baselines. This highlights the superiority of our approach to complex generative tasks. The results are given in **Appendix A.7**.

* During the rebuttal period, we took a few days to fine-tune a 770M model on different training tasks and evaluate unseen tasks to demonstrate cross-task relationships between training and evaluation tasks. Results are given in **Appendix A.8** and show tasks had stronger cross-task transfer.

All the updated changes should be reflected in the appendix of the manuscript. Additionally, we want to underscore a critical insight from our research: In large model sizes, such as 3B and 11B, the potential of MoV-style parameter-efficient MoE adapters is significantly more pronounced. Our findings highlight that our proposed MoV while maintaining a parameter budget comparable to LoRA, exhibits superior performance. Furthermore, it trains 1.3x faster than a similar parameter-sized LoRA. This demonstrates that a mixture of PEFT adapters, styled in the manner of MoV, offers a more effective and leveraged approach.

---

### Meta-Review · Area_Chair_GcUB · 2023-12-06

**Metareview:**

The paper presents a novel approach to Mixture of Experts (MoE) training, combining MoE architecture with lightweight experts to create a parameter-efficient method. The proposed Mixture of Vectors (MoV) and Mixture of LORA (MoLORA) demonstrate competitive results with full fine-tuning, while only involving 1% of the parameters. The paper is well-written and the experiments are comprehensive.

However, reviewers have raised concerns about the paper's focus on the T5 model family and its applicability to more complex tasks. There are also questions about the motivation for using multiple experts for learning, and the increase in memory consumption. The author responded with many new results, which I suggest should be included in the future version.

Despite these concerns, the paper's strengths outweigh its weaknesses. The authors' approach to MoE training is innovative and their experiments are thorough.

**Justification For Why Not Higher Score:**

While the paper presents a novel and efficient approach to MoE-PEFT training, there are a few areas that could be improved. The paper's focus on the T5 model family limits its applicability, and there are questions about its effectiveness for more complex tasks. Additionally, the reviewers raised concerns about the increase in memory consumption and the motivation for using multiple experts for learning. These areas of concern prevent the paper from receiving a higher score.

**Justification For Why Not Lower Score:**

The paper presents a novel approach to MoE-PEFT training, combining MoE architecture with lightweight experts to create a parameter-efficient method. This is a valuable contribution to the field.

The proposed Mixture of Vectors (MoV) and Mixture of LORA (MoLORA) demonstrate competitive results with full fine-tuning, while only involving 1% of the parameters.

---

### Decision · Program_Chairs · 2024-01-16

Accept (poster)